# Mitochondrial protein carboxyl-terminal alanine-threonine tailing promotes human glioblastoma growth by regulating mitochondrial function

Bei Zhang[1†], Ting Cai[1†], Esha Reddy[1], Yuanna Wu[1], Isha Mondal[2], Yinglu Tang[1], Adaeze Scholastical Gbufor[1], Jerry Wang[1‡], Yawei Shen[3,4], Qing Liu[3,4], Raymond Sun[2], Winson S Ho[2], Rongze Olivia Lu[2*], Zhihao Wu[1*]

[1]Department of Biological Sciences, Dedman College of Humanities and Sciences, Southern Methodist University, Dallas, United States; [2]Department of Neurological Surgery, University of California, San Francisco, San Francisco, United States; [3]Department of Biological Sciences, Clemson University, Clemson, United States; [4]Center for Human Genetics, Clemson University, Greenwood, United States

*For correspondence:
rongze.lu@ucsf.edu (ROL);
zhihaowu@smu.edu (ZW)

†These authors contributed
equally to this work

Present address: ‡McCombs
School of Business, University of
Texas at Austin, Austin, United
States

Competing interest: The authors
declare that no competing
interests exist.

Reviewing Editor: Rio Sugimura,
The University of Hong Kong,
Hong Kong

## eLife Assessment

Glioblastoma is among the most aggressive cancers without a cure, and its cells are characterized by high mitochondrial membrane potential. This manuscript provides **convincing** evidence that glioblastoma tumorigenesis is closely linked to mitochondrial stress. The study makes a **valuable** contribution to the field by advancing our understanding of the metabolic mechanisms driving glioblastoma and highlighting potential therapeutic targets.

**Abstract** The rapid and sustained proliferation of cancer cells necessitates increased protein production, which, along with their disrupted metabolism, elevates the likelihood of translation errors. Ribosome-associated quality control (RQC), a recently identified mechanism, mitigates ribosome collisions resulting from frequent translation stalls. However, the precise pathophysiological role of the RQC pathway in oncogenesis remains ambiguous. Our research centered on the pathogenic implications of mitochondrial stress-induced protein carboxyl-terminal alanine and threonine tailing (msiCAT-tailing), a specific RQC response to translational arrest on the outer mitochondrial membrane, in human glioblastoma multiforme (GBM). The presence of msiCAT-tailed mitochondrial proteins was observed commonly in glioblastoma stem cells (GSCs). The exogenous introduction of the mitochondrial ATP synthase F1 subunit alpha (ATP5α) protein, accompanied by artificial CAT-tail mimicking sequences, enhanced mitochondrial membrane potential ($\Delta\Psi m$) and inhibited the formation of the mitochondrial permeability transition pore (MPTP). These alterations in mitochondrial characteristics provided resistance to staurosporine (STS)-induced apoptosis in GBM cells. Consequently, msiCAT-tailing can foster cell survival and migration, whereas blocking msiCAT-tailing via genetic or pharmacological intervention can impede GBM cell overgrowth.

## Introduction

Proteins are vital to biological processes, and their overproduction is particularly crucial for rapidly proliferating cells, such as those found in cancer. To cope with this increased demand, cancer cells

extensively reform the initiation, elongation, and termination phases of their protein synthesis (*Robichaud et al., 2019*). However, heightened protein translation elevates the chance of errors (*Dever and Green, 2012*). Coupled with metabolic perturbations such as energy fluctuations and redox imbalances, the capacity to address disruptions during translation becomes indispensable. Ribosome-associated quality control (RQC) is a recently discovered suite of rescue mechanisms in eukaryotes that detect and resolve stalled, decelerated, or collided ribosomes during translation elongation or termination (*Kim and Zaher, 2022*; *Brandman et al., 2012*).

RQC is a multistep process initiated by the ZNF598/RACK1 complex, which recognizes the distinctive 40S-40S interface on collided ribosomes, triggering the ubiquitination of specific 40S subunit proteins (*Juszkiewicz and Hegde, 2017*; *Sundaramoorthy et al., 2017*). Subsequently, the ASC-1 complex separates the leading ribosome (*Hashimoto et al., 2020*; *Juszkiewicz et al., 2020*). Following this, events that transpire include: ribosomal subunit dissociation and recycling (*Shao and Hegde, 2014*), modification of the nascent peptide chains by C-terminal alanine and threonine addition (CAT-tailing) (*Shen et al., 2015*), release of CAT-tailed products from the 60S subunits by ANKZF1/VMS1 (*Verma et al., 2018*), and degradation of aberrant peptides by the Ltn1/VCP/NEMF complex (*Brandman et al., 2012*). The functional significance of CAT-tailed proteins produced during RQC remains incompletely understood. They may facilitate Ltn1-mediated ubiquitination (*Kostova et al., 2017*) and promote the degradation of defective nascent peptides by exposing lysine residues (*Lytvynenko et al., 2019*; *Sitron et al., 2020*). Nonetheless, they are also prone to forming detergent-insoluble aggregates (*Choe et al., 2016*; *Yonashiro et al., 2016*). Furthermore, contingent upon the nature of the original protein and its subcellular location, CAT-tailed proteins might possess specific, albeit currently unclear, functions. Notably, CAT-tailed proteins have been implicated in the pathogenesis of several neurodegenerative diseases, indicating a significant role in their progression (*Wu et al., 2019*; *Li et al., 2020*; *Rimal et al., 2021*).

Cancerous cells exhibit increased translation irregularities, including stop codon readthrough (*Wang and Wang, 2021*), frame-shifting (*Champagne et al., 2021*), and oxidative stress-induced ribosomal arrest (*Rubio et al., 2021*), which suggests a potential role for the RQC pathway. While CAT-tail modification of mitochondrial proteins due to compromised RQC has been noted in HeLa cells, the mechanistic involvement of RQC factors in cancer biology remains largely unexplored (*Wu et al., 2019*). Notably, the expression profile of various RQC factors (e.g. ASCC3, ABCE1, ANKZF1, and VCP) is dysregulated in cancer (*Dango et al., 2011*; *Gao et al., 2020*; *Zhou et al., 2019*; *Costantini et al., 2021*). Interestingly, RQC factors can display opposing functions in cancer development and suppression depending on specific circumstances, with some factors like ABCE1, ASCC3, and VCP suppressing cancer cell growth upon downregulation (*Dango et al., 2011*; *Gao et al., 2020*; *Costantini et al., 2021*), while others like NEMF/Clbn and ZNF598 may promote it upon inhibition (*Bi et al., 2005*; *Yang and Gupta, 2018*). This suggests a nuanced, context-dependent role for RQC components in cancer cells, influenced by both genetic and environmental factors. A recent study investigated the mechanism of ANKZF1 in mitochondrial proteostasis and its impact on glioblastoma multiforme (GBM) progression (*Li et al., 2024*). However, this study employed a nonphysiological mitochondria-targeted GFP to induce matrix proteotoxicity, leaving the role of endogenous mitochondrial proteins in this process ambiguous.

Mitochondrial stress leads to co-translational import anomalies, eliciting widespread CAT-tailing (mitochondrial stress-induced CAT-tail or msiCAT-tail) of nuclear-encoded mitochondrial proteins, including C-I30 (Complex-I 30 kDa subunit protein, NDUS3) (*Wu et al., 2019*; *Gehrke et al., 2015*). The functional ramifications of these msiCAT-tailed proteins in mitochondrial biology remain poorly elucidated. Given that CAT-tailing imparts new properties to proteins, it may contribute to the distinctive features of cancer cell mitochondria, such as hyperpolarization (*Forrest, 2015*; *Shi et al., 2019*) and resistance to drug-induced apoptosis linked to a high mitochondrial membrane potential ($\Delta\phi$m) (*Guièze et al., 2019*; *Ramamoorthy et al., 2018*; *Heerdt et al., 2006*). This membrane potential across the inner membrane of mitochondria, essential for ATP production by OXPHOS, is sustained by the electron transport chain (Complexes I to IV), which pumps protons ($H^+$) into the intermembrane space (*Zorova et al., 2018*), and ATP synthase (Complex V), which leverages this gradient (*Maria, 2020*). While numerous malignant cells exhibit reduced OXPHOS despite high energy demands (*Liberti and Locasale, 2016*), the mechanisms by which they maintain or elevate $\Delta\Psi$m remain an unresolved question (*Forrest, 2015*).

In this study, we investigated msiCAT-tailing modification on the mitochondrial ATP synthase F1 subunit alpha (ATP5α). We discerned that msiCAT-tailed ATP5α is present in GBM. The mimic short-tailed ATP5α (ATP5α-AT3 in subsequent studies) can integrate into the ATP synthase, leading to an augmented $\Delta\Psi$m and attenuated mitochondrial permeability transition pore (MPTP) assembly and opening. Consequently, msiCAT-tailed ATP5α enhances GBM cell resistance to programmed cell death induced by staurosporine (STS) and temozolomide (TMZ), thereby fostering cancer cell survival, proliferation, and migration. Conversely, impeding msiCAT-tailing diminishes cancer cell growth and resensitizes GBM cells to apoptosis. Our findings underscore the involvement of CAT-tailed mitochondrial proteins in tumorigenesis and emphasize the significance of the RQC pathway in oncobiology. These outcomes suggest that components and products of the RQC pathway may offer promising therapeutic targets for GBM.

## Results

### Presence of msiCAT-tailed proteins in glioblastoma cells

While dysregulation of individual RQC factors is documented across various cancers (e.g. adeno-carcinoma, non-small cell lung, prostate, and colon carcinomas), a comprehensive analysis of the RQC pathway in glioblastoma (GBM) has been lacking (*Dango et al., 2011*; *Gao et al., 2020*; *Zhou et al., 2019*; *Costantini et al., 2021*). Our analysis of transcriptomic data from a cohort of 153 GBM patients and 206 healthy controls, sourced from public datasets, revealed significantly elevated expression (logFC (fold change)>1; adj.P.Val <0.001) of RQC pathway genes, such as *ABCE1*, *ASCC1-3*, *RACK1*, and *VCP*, in GBM cells (*Varn et al., 2022*). Conversely, *ANKZF1* was significantly down-regulated (logFC = –0.43, adj.P.Val=0.0005) (*Figure 1A*, *Table 1*). The expression change in these genes implies activation of the RQC pathway and potential accumulation of CAT-tailed proteins in GBM. Mitochondrial stress-induced protein mitochondrial Complex-I 30 kDa (C-I 30, also known as NDUS3), an endogenous RQC substrate with msiCAT-tails, was previously identified in HeLa cells (*Wu et al., 2019*). Examination of patient-derived glioblastoma stem cells (GSCs) and normal neural stem cells (NSCs) revealed that GSCs, unlike NSCs, exhibited several msiCAT-tailed mitochondrial proteins, including NDUS3, COX4 (cytochrome *c* oxidase subunit 4), and ATP5α (ATP synthase F1 subunit alpha). Consistent with the detection of these msiCAT-tailing signals, increased NEMF (Nuclear Export Mediator Factor) levels (*Shen et al., 2015*) and decreased ANKZF1 (Ankyrin Repeat and Zinc-finger Peptidyl tRNA Hydrolase 1) expression (*Verma et al., 2018*) were observed in patient-derived GSCs (*Figure 1B*), further indicative of enhanced CAT-tailing activation, mirroring bioinformatics findings in GBM samples. A murine GBM model exhibited analogous RQC pathway alterations, with increased NEMF and decreased ANKZF1 expression in transplanted SB28 gliomas compared to normal brain tissue (*Figure 1—figure supplement 1A and B*).

The subsequent experiments were conducted using two GBM cell lines, SF268 (SF in figures) (*Rutka et al., 1987*) and GSC827 (GSC in figures) (*Kim et al., 2013*), and two control cell lines, SVG p12 (SVG in figures) and Normal Human Astrocytes E6/E7/hTERT (NHA in figures) (*Sonoda et al., 2001*). RQC protein expression analysis revealed decreased ANKZF1 and increased ABCE1, ASCC3, and NEMF expression in GSC827 and SF268 cells, consistent with findings in patient-derived GSCs (*Figure 1—figure supplement 1C*). Intriguingly, induction of CAT-tailing on a Flag-tagged β-globin reporter via a nonstop protein translation system demonstrated significantly higher CAT-tailed protein (β-globin-nonstop) production in GBM cells (*Saito et al., 2013*). This process was inhibitable by the CAT-tailing elongation inhibitor anisomycin and NEMF knockdown (sgNEMF), but not cycloheximide treatment, as evidenced by a decreased ratio of CAT-tailed (red) to non-CAT-tailed bands (green) (*Figure 1C*).

Next, to investigate the biological implications of CAT-tailing, artificial CAT-tails were introduced to mitochondrial proteins. Due to the variability in CAT-tailing, prior research simulated this process by adding alanine-threonine (AT) repeat tails to the C-terminus of mitochondrial proteins (*Wu et al., 2019*). According to recent studies, the chosen tail sequence can be stabilized by its high threonine content (*Chang et al., 2024*; *Khan et al., 2024*). ATP5α, a highly abundant mitochondrial protein with roles in cancer, was selected to study the unique functions of CAT-tailed forms (*Morgenstern et al., 2017*; *Chang et al., 2023*). siATP5α knockdown first confirmed the upper band signal in GSCs as authentic ATP5α, demonstrated by its disappearance concurrent with the main band's weakening (*Figure 1—figure supplement 2A*). Then, we confirmed that this upper band signal corresponded

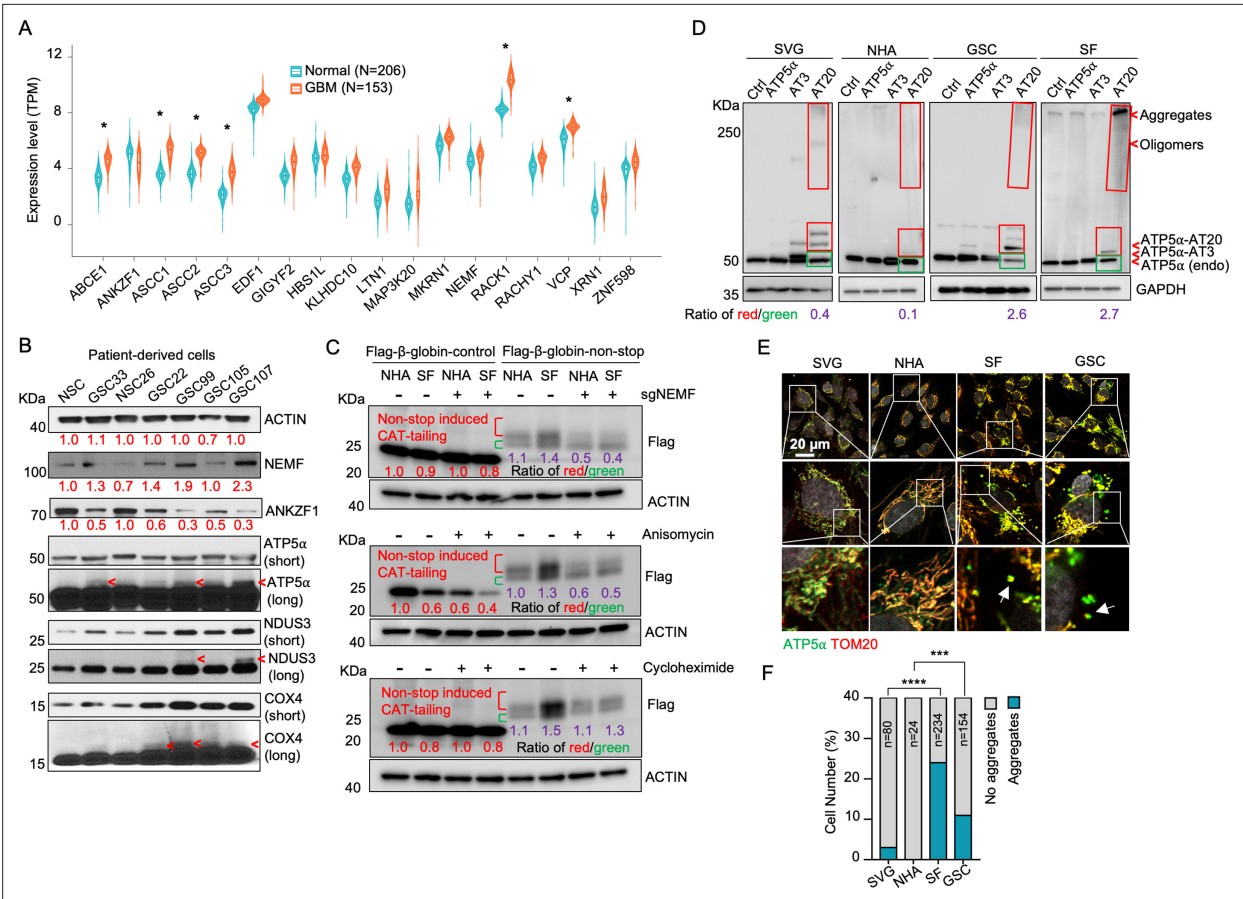

**Figure 1.** Evidence for mitochondrial stress-induced protein carboxyl-terminal alanine and threonine tailing (msiCAT-tailing) on mitochondrial proteins in glioblastoma multiforme (GBM) cells. (**A**) Ribosome-associated quality control (RQC) gene expression levels in GBM tumor tissues (n=153) compared to normal brain tissues (n=206) (unpaired Student's t-test; *, logFC (fold change)>1; adj.P.Val<0.001). (**B**) Western blot analysis of msiCAT-tailed mitochondrial proteins and RQC factors in patient-derived glioblastoma stem cells (GSCs) and control neural stem cells (NSCs), using ACTIN as the loading control. Red arrowheads indicate short CAT-tailed mitochondrial proteins; 'short' and 'long' refer to exposure time; the red numbers represent fold changes compared to controls (NSC). (**C**) Western blot of 5×FLAG-tagged β-globin reporter proteins in GBM and control cells, showing more CAT-tailed proteins in GBM cells, using ACTIN as the loading control. The red numbers represent fold changes compared to controls (NHA without any treatment); the purple numbers represent the ratio of red (CAT-tailed) to green (non-CAT-tailed) sections. (**D**) Western blot of overexpressed ATP5α-AT3 and ATP5α-AT20 in GBM and control cells, using GAPDH as the loading control; arrowheads indicate endogenous ATP5α, ATP5α-AT3, ATP5α-AT20, and oligomers/aggregates of msiCAT-tailed ATP5α proteins. The purple numbers represent the ratio of red (exogenous) to green (endogenous) sections. (**E**) Immunofluorescence staining shows endogenous ATP5α protein aggregates in GBM cells, with TOM20 (red) as a mitochondrial marker. White arrows indicate ATP5α protein aggregates. (**F**) Quantification of E (n=3; chi-squared test; ***, p<0.001; ****, p<0.0001); the total number of cells counted is indicated in the columns.

The online version of this article includes the following source data and figure supplement(s) for figure 1:

**Source data 1.** PDF file containing original western blots for *Figure 1B, C, and D*, indicating the relevant bands and treatments.

**Source data 2.** Original files for western blot analysis shown in *Figure 1B, C, and D*.

**Source data 3.** Numerical source data shown in *Figure 1A, F*.

**Figure supplement 1.** Ribosome-associated quality control (RQC) pathway activity in glioblastoma multiforme (GBM) cells.

**Figure supplement 1—source data 1.** PDF file containing original western blots for *Figure 1—figure supplement 1C*, indicating the relevant bands and treatments.

**Figure supplement 1—source data 2.** Original files for western blot analysis shown in *Figure 1—figure supplement 1C*.

**Figure supplement 1—source data 3.** Numerical source data shown in *Figure 1—figure supplement 1B*.

**Figure supplement 2.** AT repeat sequences mimicking CAT-tails induce protein aggregates in cells.

**Figure supplement 2—source data 1.** PDF file containing original western blots for *Figure 1—figure supplement 2A, B, and E*, indicating the relevant bands and treatments.

*Figure 1 continued on next page*

*Figure 1 continued*

**Figure supplement 2—source data 2.** Original files for western blot analysis shown in *Figure 1—figure supplement 2A, B, and E*.

**Figure supplement 2—source data 3.** Numerical source data shown in *Figure 1—figure supplement 2D, G*.

**Figure supplement 3.** Aggregation of CAT-tailed mitochondrial proteins observed in vivo.

**Figure supplement 3—source data 1.** Numerical source data shown in *Figure 1—figure supplement 3B, D*.

to changes in CAT-tailing, which could be effectively inhibited by NEMF knockdown and anisomycin treatment (*Figure 1—figure supplement 2B*). Due to the indistinct nature of the endogenous msiCAT-tailed ATP5α signal, exogenously expressed Flag-ATP5α was utilized here.

To investigate the potential new function provided by CAT-tailed proteins, control (SVG and NHA) and GBM (SF268 and GSC827) cell lines overexpressed ATP5α with three (ATP5α-AT3) or twenty (ATP5α-AT20) AT repeats. Consistent with earlier findings, only the long-tailed ATP5α-AT20 exhibited posttranslational modifications and detergent-resistant insoluble aggregates, appearing as slower migrating bands and a high-molecular-weight smear in protein electrophoresis (*Figure 1D*). Based on comparing exogenously expressed (indicated by red boxes) to endogenous proteins (indicated by green boxes), GBM cell lines (GSC827, SF268) showed increased accumulation of ATP5α-AT20 compared to control cells (SVG, NHA). This accumulation may occur due to increased stability and reduced degradation of long-tailed proteins, a malfunctioning protein quality control system, enhanced cellular tolerance to protein accumulation, or a combination of these factors. Subcellular localization analysis showed that the short AT tail (AT3) did not significantly alter ATP5α's mitochondrial localization, similar to the tailless protein. However, a significant portion of ATP5α-AT20 was found in the cytoplasm near mitochondria, forming protein aggregates, with the highest proportion in highly malignant GSCs (*Figure 1—figure supplement 2C and D*). Notably, poly-glycine-serine tails (short, GS3, and long, GS20) did not induce insoluble protein aggregation or intracellular punctate

**Table 1.** Differential expression analysis of ribosome-associated quality control (RQC) genes in glioblastoma multiforme (GBM) patients compared to healthy controls.

| Gene | logFC | AveExpr | t | P.Value | adj.P.Val |
|------|-------|---------|---|---------|-----------|
| *RACK1* | 2.224548565 | 9.048113333 | 19.24641934 | 2.62E-57 | 6.69E-56 |
| *ASCC3* | 1.738216567 | 2.717246389 | 19.27211713 | 2.05E-57 | 5.27E-56 |
| *ASCC1* | 1.689584768 | 4.257915556 | 17.8774401 | 1.20E-51 | 2.18E-50 |
| *ASCC2* | 1.471467207 | 4.153399167 | 15.9118075 | 1.43E-43 | 1.68E-42 |
| *ABCE1* | 1.32826428 | 3.81695 | 13.79248369 | 4.69E-35 | 3.60E-34 |
| *VCP* | 1.050066326 | 6.321021667 | 9.828944092 | 2.33E-20 | 9.31E-20 |
| *GIGYF2* | 0.985695112 | 3.786421389 | 10.25440005 | 8.00E-22 | 3.41E-21 |
| *MAP3K20* | 0.962218073 | 1.863793889 | 8.711292467 | 1.10E-16 | 3.75E-16 |
| *PELO* | 0.92860628 | 2.2885625 | 10.60122263 | 4.85E-23 | 2.18E-22 |
| *KLHDC10* | 0.854921284 | 3.492322222 | 9.064976509 | 8.08E-18 | 2.88E-17 |
| *EDF1* | 0.82091202 | 8.444505 | 7.278600017 | 2.12E-12 | 5.94E-12 |
| *XRN1* | 0.809119864 | 1.518371111 | 8.54020086 | 3.80E-16 | 1.26E-15 |
| *LTN1* | 0.786409776 | 1.9716 | 9.962815742 | 8.14E-21 | 3.32E-20 |
| *MKRN1* | 0.769369764 | 5.745359444 | 7.400071791 | 9.65E-13 | 2.74E-12 |
| *RCHY1* | 0.652647968 | 4.276126111 | 6.840213538 | 3.40E-11 | 8.93E-11 |
| *ZNF598* | 0.62380412 | 4.006663611 | 6.043582081 | 3.76E-09 | 8.90E-09 |
| *HBS1L* | 0.291107388 | 4.701389722 | 2.72853772 | 0.006672805 | 0.010370549 |
| *NEMF* | 0.194631373 | 4.566962778 | 2.575063266 | 0.010419695 | 0.015855894 |
| *ANKZF1* | –0.436070986 | 4.620298333 | –3.65005718 | 0.000300886 | 0.000525859 |

distribution (*Figure 1—figure supplement 2E–G*), highlighting the importance of specific amino acid composition.

Importantly, in GBM cells, both exogenous tailed proteins and the endogenous ATP5α formed clusters attached to the outer mitochondrial membrane (*Figure 1E and F*). Similar aggregate formation in GBM cells was also observed with other mitochondrial proteins, such as NDUS3 (*Figure 1—figure supplement 3A and B*). Furthermore, we examined the mouse GBM models. Akin to in vitro culture, ATP5α in transplanted SB28 glioma formed more punctate signals and did not always colocalize with the mitochondrial marker TOM20 (*Figure 1—figure supplement 3C–E*). These findings collectively indicate a disruption of the RQC pathway, leading to the presence of msiCAT-tailed proteins in GBM cells.

## msiCAT-tailed ATP5α elevates mitochondrial membrane potential (ΔΨm)

Some cancer cells exhibit altered mitochondrial physiology, maintaining or increasing mitochondrial membrane potential (ΔΨm) despite reduced respiration. This was observed in patient-derived GSCs, which demonstrated higher ΔΨm but lower ATP production than control NSCs (*Figure 2A and B*). Similarly, GBM cell lines, GSC827 and SF268, displayed comparable or higher ΔΨm and lower ATP levels relative to the control NHA cell line (*Sonoda et al., 2001*; *Figure 2—figure supplement 1A–C*). Genetic inhibition of msiCAT-tailing, via NEMF knockdown (sgNEMF) or ANZKF1 overexpression (oeANZKF1) (*Figure 2—figure supplement 1D*), as well as pharmacological inhibition by anisomycin treatment, effectively reduced ΔΨm in GBM cells but not in NHA cells (*Figure 2C and D*).

Our next investigation of msiCAT tail proteins revealed their impact on mitochondrial function. Expression of Flag-tagged ATP5α-AT3 and ATP5α-AT20 in GBM and control cell lines elevated ΔΨm specifically in GBM cells (*Figure 2E*). Overexpression of ATP5α-GS3 and ATP5α-GS20 did not exhibit this effect (*Figure 2—figure supplement 1E*). To our surprise, even with suppressed endogenous CAT-tailing through sgNEMF and oeANZKF1 in GSCs, the introduced AT3 and AT20 proteins could still effectively elevate ΔΨm (*Figure 2F and G*). This finding suggests that CAT-tailing of ATP5α may be a significant contributor to the observed mitochondrial phenotype (*Figure 2G*). Blue Native polyacrylamide gel electrophoresis (BN-PAGE) illustrated distinct effects based on CAT-tail length. ATP5α-AT3 integrated into the mitochondrial respiratory chain complex, whereas ATP5α-AT20 formed high-molecular-weight complexes or remained as monomers (*Figure 2H*). In mitochondrial physiological activity assays using the Agilent Cell Mitochondrial Stress Test, the oxygen consumption rate (OCR) was directly measured to assess mitochondrial respiration. Our findings indicate that expressing both ATP5α-AT3 and ATP5α-AT20 negatively impacted mitochondrial oxidative phosphorylation. This impairment leads to a reduction in ATP synthesis, basal respiration, and maximal respiration rates (*Figure 2I–L*). These data suggest that both short and long tails on ATP5α proteins influence mitochondrial function, although potentially through different mechanisms. Short CAT-tails may directly act on ATP synthase function and thus affect the respiratory chain complex, while long CAT-tails form protein aggregates, causing mitochondrial proteostasis stress and thus indirectly affecting mitochondrial respiration (*Wu et al., 2019*; *Li et al., 2024*). This differential impact of CAT-tail length suggests a nuanced regulation of mitochondrial function mediated by ATP5α modifications.

## msiCAT-tailing influences MPTP dynamics

Beyond its traditionally recognized role in ATP production, the $F_1F_0$ ATP synthase has garnered increasing attention as a potential structural component of the MPTP complex (*Alavian et al., 2014*; *Giorgio et al., 2013*; *Bonora et al., 2013*). Given the possibility that CAT-tailed proteins like ATP5α might modulate MPTP function, this investigation sought to elucidate the mechanism by which msiCAT-tailing modulates MPTP dynamics (open-close state). Comparative analyses conducted in GBM and control cells revealed that MPTP in GSC827 cells predominantly exists in a closed conformation, indicated by strong Calcein signals. Notably, the treatment of anisomycin, a pharmacological CAT-tailing inhibitor, effectively induced MPTP opening in GSC827 cells, as indicated by decreased Calcein signals (*Figure 3A and B*). This effect was concomitant with the diminished aggregation of endogenous ATP5α (*Figure 3E and F*). Furthermore, corroborative evidence was obtained through genetic manipulation. Specifically, genetic inhibition of CAT-tailing via NEMF knockdown (sgNEMF) resulted in a similar decrease in Calcein signaling and a reduction in ATP5α accumulation (*Figure 3C, D, G, and H*), aligning with the results obtained using anisomycin. In

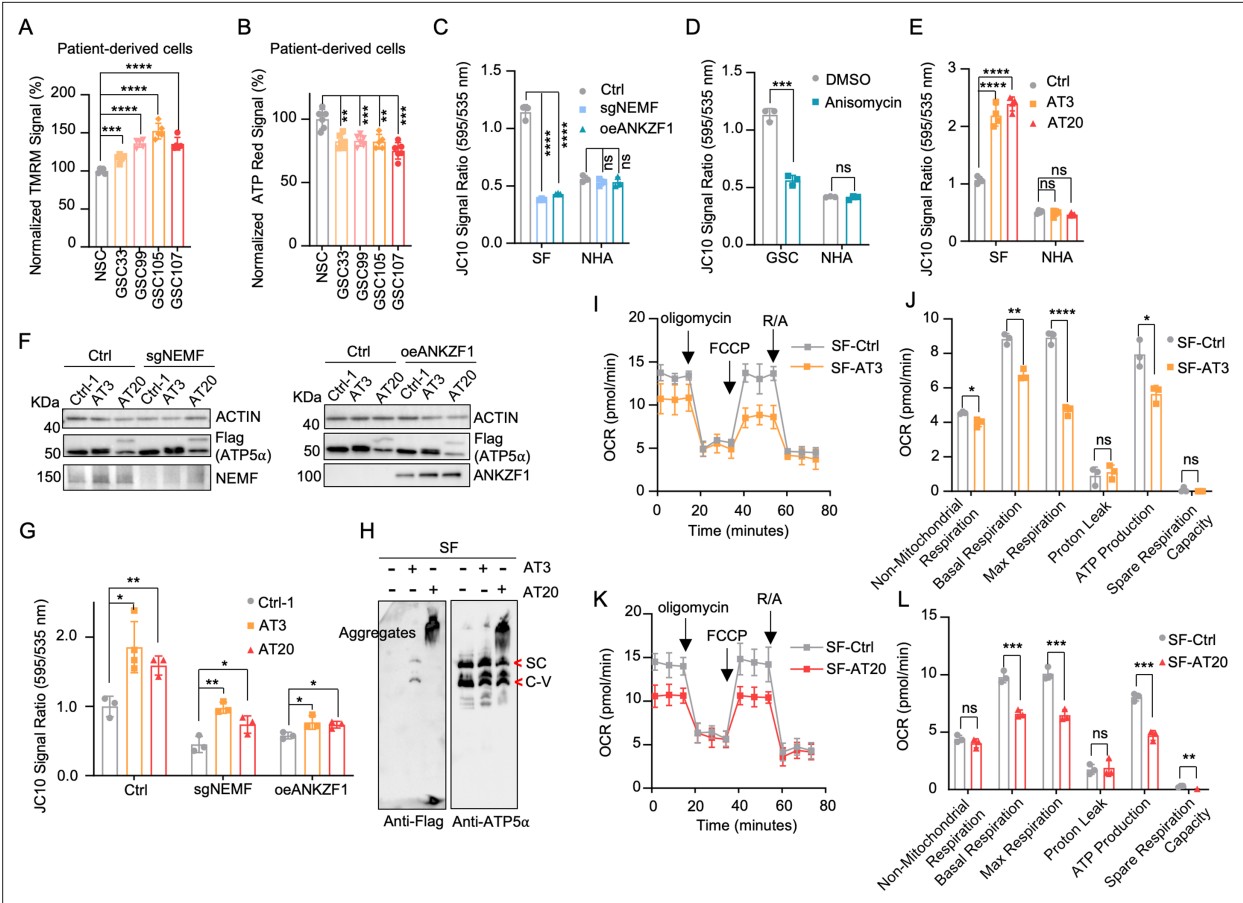

**Figure 2.** Impact of msiCAT-tailed ATP5α proteins on mitochondrial functions in glioblastoma multiforme (GBM) cells. (**A**) TMRM staining shows a high mitochondrial membrane potential in patient-derived glioblastoma stem cells (GSCs) (n=3; unpaired Student's t-test; ***, p<0.001; ****, p<0.0001). (**B**) ATP measurement shows a low mitochondrial ATP production in patient-derived GSCs (n=3; unpaired Student's t-test; **, p<0.01; ***, p<0.001). (**C, D**) JC-10 staining reveals a reduced mitochondrial membrane potential in GBM cells, but not in NHA control cells, upon both genetic (**C**) and pharmacological (**D**) inhibition of the mitochondrial stress-induced protein carboxyl-terminal alanine and threonine tailing (msiCAT-tailing) pathway (n=3; unpaired Student's t-test; ***, p<0.001; ****, p<0.0001; ns, not significant). (**E**) JC-10 staining reveals an increased mitochondrial membrane potential in GBM cells, but not in control cells, upon overexpression of ATP5α-AT3 and ATP5α-AT20 (n=3; unpaired Student's t-test; ****, p<0.0001; ns, not significant). (**F**) Western blot of FLAG-tagged ATP5α, NEMF, and ANKZF1 in GBM cells and control cells, using ACTIN as the loading control. (**G**) JC-10 staining reveals an increased mitochondrial membrane potential in GBM cells, but not in NHA control cells, upon overexpression of ATP5α-AT3 and ATP5α-AT20 with concurrent genetic inhibition of the endogenous msiCAT-tailing pathway (n=3; unpaired Student's t-test; *, p<0.05; **, p<0.01). (**H**) Blue Native polyacrylamide gel electrophoresis (BN-PAGE) western blot of ATP5α and Flag shows that ATP5α-AT3 is incorporated into the mitochondrial Complex-V (ATP synthase), while ATP5α-AT20 forms high-molecular-weight protein aggregates in GBM cells. SC: respiratory supercomplex; C-V: Complex-V/ATP synthase. (**I, K**) Oxygen consumption rate (OCR) data indicate a reduction in mitochondrial oxygen consumption in SF268 cells expressing ATP5α-AT3 and ATP5α-AT20. Oligomycin (1.5 μM), FCCP (1.0 μM), and rotenone/antimycin A (R/A, 0.5 μM) were sequentially added. (**J, L**) Statistics of mitochondrial respiration parameters in (**I, K**), including non-mitochondrial respiration, basal respiration, maximum respiration, spare respiration, proton leaks, and ATP production (n=3; unpaired Student's t-test; *, p<0.05; **, p<0.01; ***, p<0.001; ****, p<0.0001; ns, not significant).

The online version of this article includes the following source data and figure supplement(s) for figure 2:

**Source data 1.** PDF file containing original western blots for *Figure 2F and H*, indicating the relevant bands and treatments.

**Source data 2.** Original files for western blot analysis shown in *Figure 2F and H*.

**Source data 3.** Numerical source data shown in *Figure 2A–L*.

**Figure supplement 1.** Aberrant mitochondrial function in glioblastoma multiforme (GBM) cells.

**Figure supplement 1—source data 1.** PDF file containing original western blots for *Figure 2—figure supplement 1D*, indicating the relevant bands and treatments.

**Figure supplement 1—source data 2.** Original files for western blot analysis shown in *Figure 2—figure supplement 1D*.

**Figure supplement 1—source data 3.** Numerical source data shown in *Figure 2—figure supplement 1A, C and E*.

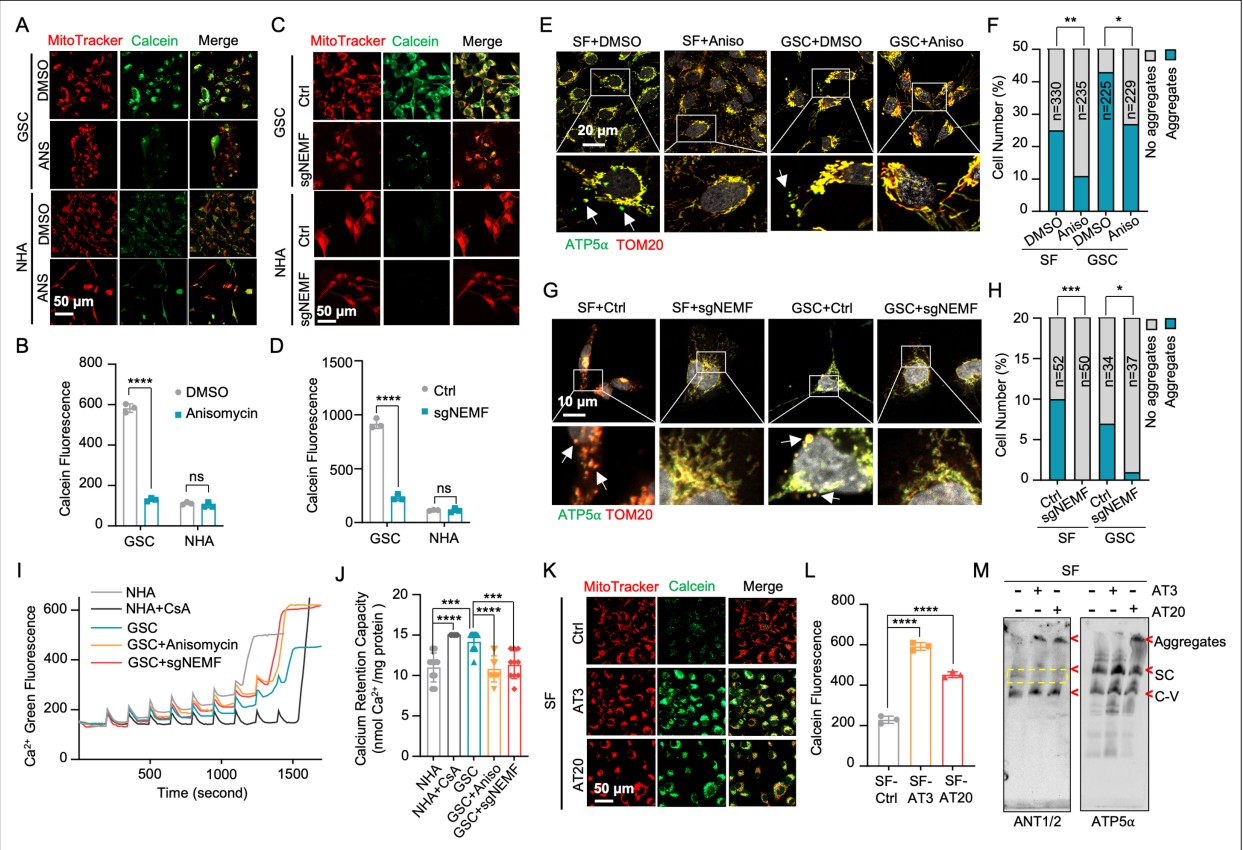

**Figure 3.** Mitochondrial stress-induced protein carboxyl-terminal alanine and threonine tailing (msiCAT-tailing) product regulates mitochondrial permeability transition pore (MPTP) status in glioblastoma multiforme (GBM) cells. (**A, C**) MPTP activity assay shows reduced MPTP opening in glioblastoma stem cells (GSCs) compared to NHA (control) cells. Pharmacological (A, anisomycin 200 nM) and genetic (sgNEMF) inhibition of CAT-tailing reverse it. (**B, D**) Quantification of (**A, C**) (n=3; unpaired Student's t-test; ****, p<0.0001; ns, not significant). (**E, G**) Immunofluorescence staining shows that anisomycin treatment (**E**) and sgNEMF (**G**) inhibit endogenous ATP5α protein aggregation in GBM cells, using TOM20 (red) as a mitochondrial marker. (**F, H**) Quantification of (**E, G**) (n=3; chi-squared test; *, p<0.05; **, p<0.01; ***, p<0.001); the total number of cells counted is indicated in the columns. (**I**) The calcium retention capacity (CRC) assay of isolated mitochondria, measured using the Calcium Green-5N dye, reveals a significantly higher CRC in GBM cells compared to control NHA cells. CsA (Cyclosporin A, MPTP inhibitor) serves as a positive control. (**J**) Statistic of (**I**) shows attenuated CRC in mitochondria pre-treated with anisomycin or with sgNEMF (n=10; unpaired Student's t-test; ***, p<0.001; ****, p<0.0001). (**K**) MPTP activity assay shows that ectopic expression of ATP5α-AT3 and ATP5α-AT20 inhibits MPTP opening in GBM cells. (**L**) Quantification of (**K**) (n=3; unpaired Student's t-test; ****, p<0.0001). (**M**) Blue Native polyacrylamide gel electrophoresis (BN-PAGE) western blot shows that ATP5α-AT3 and ATP5α-AT20 expression alters ANT1/2 protein patterns in GBM cells, resulting in a missing band (circled in yellow dashed line) and formation of high-molecular-weight aggregates. SC: respiratory supercomplex; C-V: Complex V/ATP synthase.

The online version of this article includes the following source data and figure supplement(s) for figure 3:

**Source data 1.** PDF file containing original western blots for *Figure 3M*, indicating the relevant bands and treatments.

**Source data 2.** Original files for western blot analysis shown in *Figure 3M*.

**Source data 3.** Numerical source data shown in *Figure 3B, D, F, H, I, J, and L*.

**Figure supplement 1.** Cycloheximide does not impact mitochondrial functions.

**Figure supplement 1—source data 1.** Numerical source data shown in *Figure 3—figure supplement 1B, D, and E*.

**Figure supplement 2.** The CAT-tailed ATP5α variant has no interaction with mitochondrial permeability transition pore (MPTP) proteins.

**Figure supplement 2—source data 1.** PDF file containing original western blots for *Figure 3—figure supplement 2C and D*, indicating the relevant bands and treatments.

**Figure supplement 2—source data 2.** Original files for western blot analysis shown in *Figure 3—figure supplement 2C and D*.

**Figure supplement 2—source data 3.** Numerical source data shown in *Figure 3—figure supplement 2A, B*.

contrast, treatment with cycloheximide, a general translation inhibitor, did not significantly alter Calcein or ATP5α aggregation signals (*Figure 3—figure supplement 1A–D*), suggesting that nonspecific translation inhibition does not impact the mitochondrial MPTP state. The crucial role of CAT-tail modifications on ATP5α in modulating MPTP status was further substantiated by the observation that overexpression of artificially synthesized AT repeat tails (AT3 and AT20) restored Calcein signals despite the inhibition of endogenous CAT-tailing (*Figure 3—figure supplement 1E*).

The MPTP is recognized to participate in the transient efflux of protons, calcium ions ($Ca^{2+}$), and other signaling molecules from the mitochondrial matrix during brief opening episodes (*Ichas et al., 1997*). To quantitatively evaluate the MPTP open/closed state, the mitochondrial $Ca^{2+}$ retention capacity (CRC) assay was employed, which measures the amount of $Ca^{2+}$ required to elicit MPTP opening. Our results revealed that GSC827 cells exhibited a greater CRC value than NHA cells. Pre-treatment with anisomycin or knockdown of NEMF (sgNEMF) significantly decreased the CRC in GBM cells, indicating MPTP opening upon the loss of CAT-tailed proteins (*Figure 3I and J*). Consistent with Calcein staining results (*Figure 3—figure supplement 1A and B*), cycloheximide treatment did not substantially alter CRC measurements (*Figure 3—figure supplement 2A and B*). Conversely, enhancing CAT-tailing (e.g. via oeNEMF and siANKZF1) led to an increase in CRC (*Figure 3—figure supplement 2A and B*), although this effect was less pronounced in GSCs, potentially due to their inherently active CAT-tailing and closed MPTP.

To further investigate the impact of specific AT repeat tails on MPTP opening, artificial AT repeat tails on ATP5α were introduced into GBM cells. It was found that the short AT tail (AT3) inhibited MPTP opening, while the long AT tail (AT20) displayed a weaker effect (*Figure 3K and L*), potentially due to their different integration into ATP synthase (*Figure 2G*). Complex co-immunoprecipitation (co-IP) assay did not detect direct interactions between ATP5α with AT3 or AT20 tails and MPTP components cyclophilin D (CypD) and adenine nucleotide translocator 2 (ANT2) (*Figure 3—figure supplement 2C*). However, CypD expression was reduced upon ectopic expression of ATP5α-AT3 and ATP5α-AT20, suggesting decreased MPTP formation (*Figure 3—figure supplement 2D*). Intriguingly, BN-PAGE analysis revealed that both ATP5α-AT3 and ATP5α-AT20 altered ANT1/2-containing complexes, with expected bands disappearing (indicated by *) and aggregates forming (at the top), supporting the notion that ATP synthase is integrated into the MPTP supercomplex due to the spatial proximity of the ANT1/2 complex and ATP synthase (*Figure 3M*). In conclusion, msiCAT-tailed ATP5α proteins, particularly those with short AT3 tails, are integrated into ATP synthase and have a substantial influence on modulating MPTP status.

## msiCAT-tailing boosts GBM cell migration and resistance to apoptosis

The elevated mitochondrial membrane potential ($\Delta\Psi m$) and constricted MPTP resulting from msiCAT-tailed ATP5α and other mitochondrial proteins may enhance cellular stress resilience. We first investigated how the msiCAT-tailing mechanism affects GBM cells at the cellular level. MTT assays (*Stockert et al., 2018*) revealed that overexpressing short (AT3) and long (AT20) AT repeat tails, fused to ATP5α, significantly improved GBM cell viability, but not that of NHA cells (*Figure 4A and B*). However, short (GS3) and long (GS20) GS repeat tails did not affect GBM cell viability (*Figure 4—figure supplement 1A*). In addition, in vitro transwell migration assays (*Justus et al., 2014*) and wound healing assays (*Grada et al., 2017*) showed that GBM cells overexpressing AT repeat-tailed ATP5α exhibited increased cell invasion and accelerated wound healing, indicating enhanced cell migration (*Figure 4C and D*, *Figure 4—figure supplement 1B and C*). Notably, neither ATP5α alone nor GS repeat-tailed proteins showed comparable changes (*Figure 4—figure supplement 1D and E*). Furthermore, overexpressing AT3- and AT20-tailed proteins effectively conferred phenotypes associated with increased GBM cell activity, such as enhanced survival and migration, even with inhibited endogenous CAT-tailing machinery activity (e.g. sgNEMF and oeANKZF1) (*Figure 4E–G*). It is worth noting that ANKZF1 knockdown in U87 and U251 cell lines can cause aberrant mitoGFP accumulation, possibly reducing cellular adaptability (*Li et al., 2024*), suggesting varying mitochondrial adaptability to proteostasis stress across cell lines. Supporting this, initial experiments showed that mild expression of ATP5α-AT3 and ATP5α-AT20 did not induce strong mitochondrial proteotoxic responses, as evidenced by the lack of significant upregulation in *LONP1*, *mtHSP70*, and *HSP60* mRNA levels (*Figure 4—figure supplement 1F*).

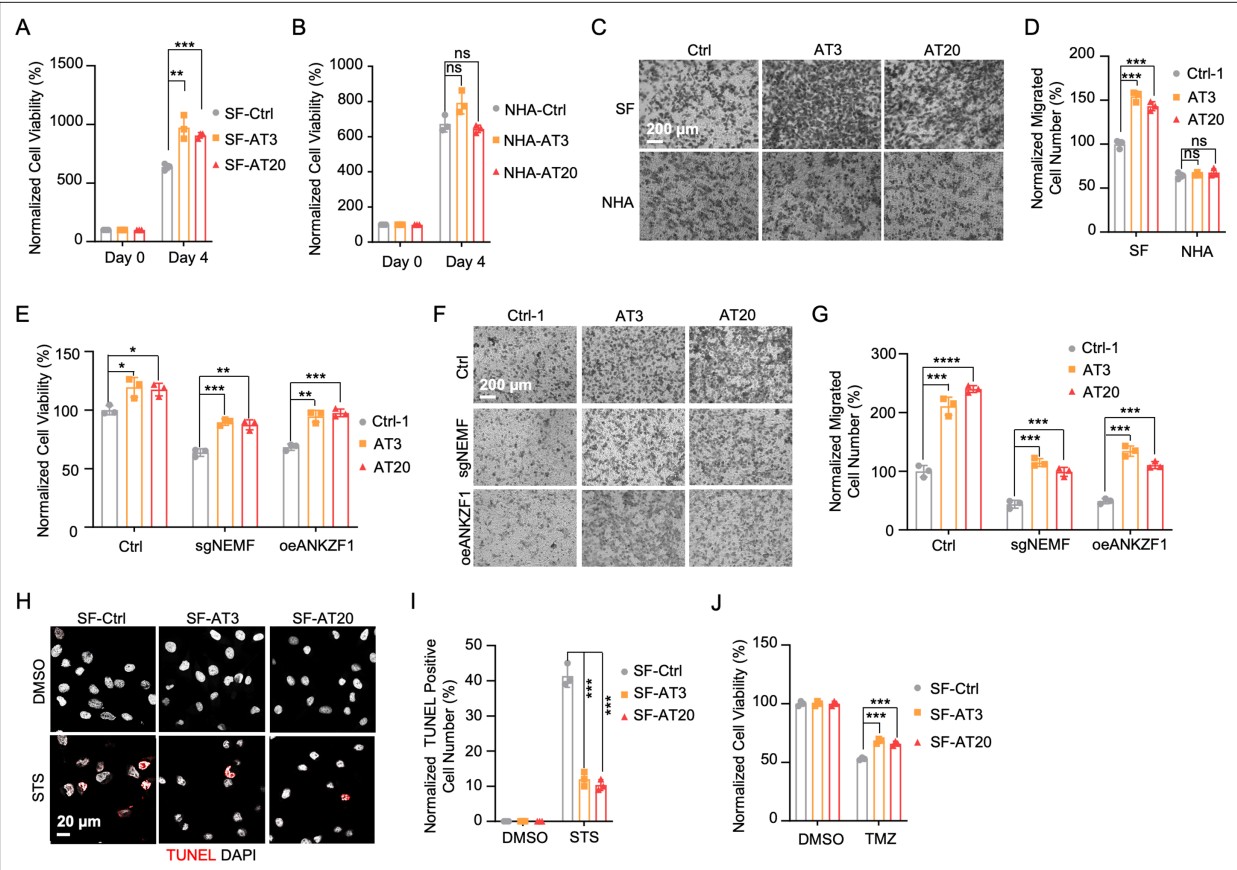

**Figure 4.** msiCAT-tailed ATP5α protein promotes glioblastoma multiforme (GBM) progression. (**A**) MTT assay indicates increased proliferation caused by ATP5α-AT3 and ATP5α-AT20 expression in GBM cells (n=3; unpaired Student's t-test; **, p<0.01; ***, p<0.001). (**B**) MTT assay indicates no change in proliferation caused by ATP5α-AT3 and ATP5α-AT20 expression in NHA cells (n=3; unpaired Student's t-test; ns, not significant). (**C**) Transwell assay reveals enhanced migration induced by ATP5α-AT3 and ATP5α-AT20 expression in GBM (SF) cells but not in control (NHA) cells. (**D**) Quantification of (**C**) shows the number of migrated cells (n=3; unpaired Student's t-test; ***, p<0.001; ns, not significant). (**E**) MTT assay indicates an increased proliferation in GBM cells, upon overexpression of ATP5α-AT3 and ATP5-AT20 with concurrent genetic inhibition of the endogenous mitochondrial stress-induced protein carboxyl-terminal alanine and threonine tailing (msiCAT-tailing) pathway (n=3; unpaired Student's t-test; *, p<0.05; **, p<0.01). (**F**) Transwell assay reveals enhanced migration upon overexpression of ATP5α-AT3 and ATP5α-AT20 with concurrent genetic inhibition of the endogenous msiCAT-tailing pathway. (**G**) Quantification of (**F**) shows the number of migrated cells (n=3; unpaired Student's t-test; ***, p<0.001; ****, p<0.0001). (**H**) TUNEL staining shows that staurosporine (STS, 1 μM)-induced apoptosis is attenuated by ATP5α-AT3 and ATP5α-AT20 expression in GBM cells, using TUNEL-Cy3 as an apoptotic cell indicator and DAPI as a nucleus indicator. (**I**) Quantification of (**H**) shows the percentage of TUNEL-positive cells in the population (n=3; unpaired Student's t-test; ***, p<0.001), using DMSO as the vehicle control. (**J**) MTT assay indicates an enhanced resistance to temozolomide (TMZ, 150 μM) induced by ATP5α-AT3 and ATP5α-AT20 expression. The TMZ-treated/SF-Ctrl group is used as the control (n=3; unpaired Student's t-test; ***, p<0.001).

The online version of this article includes the following source data and figure supplement(s) for figure 4:

**Source data 1.** Numerical source data shown in *Figure 4A, B, D, E, G, I, and J*.

**Figure supplement 1.** Effect of GS repeat tails on glioblastoma multiforme (GBM) proliferation and migration.

**Figure supplement 1—source data 1.** Numerical source data shown in *Figure 4—figure supplement 1A, C, E, and F*.

**Figure supplement 2.** Glioblastoma multiforme (GBM) cells exhibit increased resistance to apoptosis.

**Figure supplement 2—source data 1.** PDF file containing original western blots for *Figure 4—figure supplement 2C*, indicating the relevant bands and treatments.

**Figure supplement 2—source data 2.** Original files for western blot analysis shown in *Figure 4—figure supplement 2C*.

**Figure supplement 2—source data 3.** Numerical source data shown in *Figure 4—figure supplement 2B, E, and G*.

GBM cells exhibit increased resistance to STS-induced apoptosis, supported by fewer TUNEL-positive cells (*Figure 4—figure supplement 2A and B*) and markedly diminished PARP-1 (poly ADP-ribose polymerase) cleavage (*Figure 4—figure supplement 2C*), a marker of AIF-mediated apoptosis (*Mashimo et al., 2021*). To investigate the role of CAT-tailed ATP5α proteins in this resistance, we overexpressed proteins with mimetic tails in GBM cells. Overexpression of both short tail (ATP5α-AT3) and long tail (ATP5α-AT20) significantly enhanced resistance to STS-induced apoptosis, as shown by TUNEL staining (*Figure 4H and I*) and flow cytometry (*Figure 4—figure supplement 2D and E*), indicating a strong link between protein CAT-tailing and tumorigenesis. In contrast, control short (GS3) and long (GS20) GS tails failed to confer such resistance (*Figure 4—figure supplement 2F and G*). Consistent with these findings, overexpression of artificial CAT-tailed ATP5α proteins also increased the resistance of GBM cells to TMZ-induced apoptosis (*Figure 4J*). Taken together, these results suggest that RQC-induced CAT-tailing on ATP5α protein plays a role in GBM resistance to drug-induced apoptosis.

## RQC pathway inhibition hinders GBM cell progression

Prior research indicates the RQC pathway-mediated msiCAT-tailing plays an important role in GBM progression, suggesting it as a potential therapeutic target. To explore this, patient-derived GSC lines were treated with anisomycin, an inhibitor of CAT-tailing. GSC lines displayed higher sensitivity to anisomycin than normal NSCs (*Figure 5A*). Similarly, genetic inhibition of the RQC pathway via NEMF knockdown (sgNEMF) or ANKZF1 overexpression (oeANZKF1) in the SF268 GBM cell line also suppressed GBM growth (*Figure 5B*). Notably, control NHA cell proliferation was also inhibited by these genetic changes, indicating the broad significance of NEMF and ANKZF1 in cell proliferation (*Figure 5C*). The RQC pathway appears to have a more pronounced effect on GBM cell migration. In in vitro transwell assays, sgNEMF or oeANZKF1 notably decreased GBM cell migration without affecting NHA cells (*Figure 5D and E*). Consistently, anisomycin treatment impaired GSC migration, but not NHA cell migration (*Figure 5F and G*).

Further investigation revealed the RQC pathway's involvement in GBM cell anti-apoptosis, with initial findings pointing to alterations in mitochondrial functions. Prior studies demonstrated that genetic or pharmacological inhibition of the RQC pathway led to a significant decrease in GBM mitochondrial membrane potential ($\Delta\Psi$m) (*Figure 2C and D*). In GSCs, anisomycin treatment promoted MPTP opening, an effect not seen in NHA cells (*Figure 3A–D*). Consequently, GBM cell lines with genetically or pharmacologically inhibited RQC pathways were more susceptible to STS-induced apoptosis, evidenced by elevated executioner caspase 3/7 activity (*Figure 5—figure supplement 1A*), enhanced PARP-1 cleavage (*Figure 5—figure supplement 1B and C*), increased TUNEL staining (*Figure 5H–K*), and flow cytometry analysis (*Figure 5—figure supplement 1D–G*). Notably, general translation inhibition using cycloheximide did not elicit the same apoptotic response (*Figure 5—figure supplement 1A, D, and E*). Finally, the RQC pathway was also implicated in TMZ-induced cell death. Combining anisomycin with TMZ significantly reduced GBM cell survival (*Figure 5L*) and effectively inhibited GSC spheroid growth (*Figure 5M and N*). In summary, the RQC pathway plays a critical role in multiple aspects of GBM progression, including proliferation, migration, and survival under apoptotic stress.

## Discussion

The RQC pathway plays a crucial role in managing aberrant proteins produced during translation. This study focused on understanding the consequences of RQC-mediated modification, specifically the addition of msiCAT tails, on mitochondrial proteins such as ATP5α in GBM cells. The findings reveal that GBM cells harboring msiCAT-modified ATP5α exhibit a unique metabolic profile. Despite a reduction in ATP synthesis, these cells maintain their mitochondrial membrane potential ($\Delta\Psi$m), a key factor for cellular function and survival. Furthermore, they demonstrate enhanced cell survival and motility, characteristics associated with increased tumor invasiveness and metastasis. Notably, the presence of msiCAT-modified ATP5α confers resistance to apoptosis triggered by STS, potentially by modulating the MPTP, a critical regulator of cell death pathways, as illustrated in *Figure 6*. These identified traits contribute to an increased aggressiveness of tumors, suggesting that the RQC pathway plays a critical role in cancer cell survival and proliferation. Encouragingly, a recent study also demonstrated the RQC

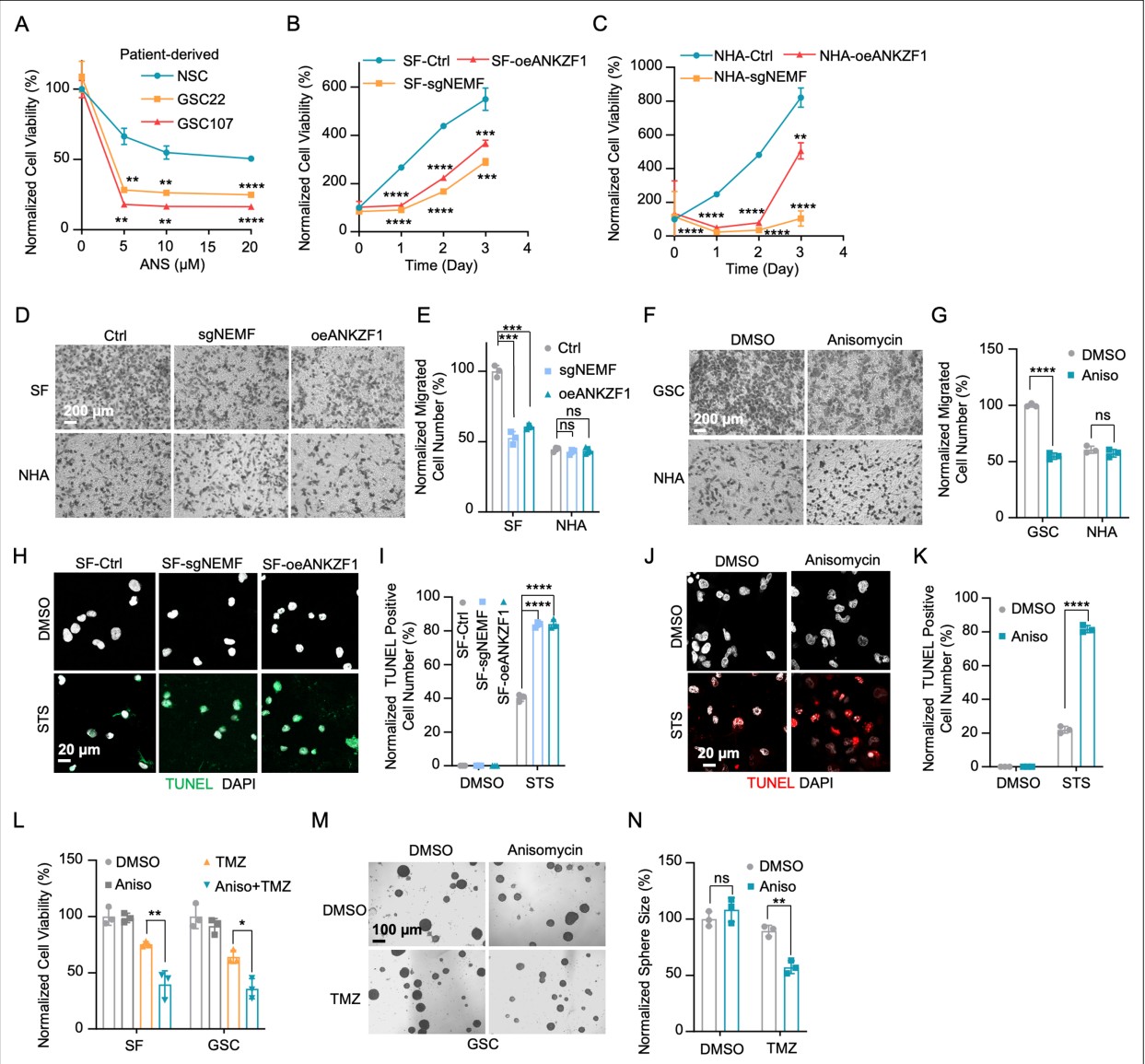

**Figure 5.** Inhibition of mitochondrial stress-induced protein carboxyl-terminal alanine and threonine tailing (msiCAT-tailing) impedes glioblastoma multiforme (GBM) progression. (**A**) Cell viability assay shows greater sensitivity to anisomycin treatment in patient-derived glioblastoma stem cells (GSCs) than control neural stem cells (NSCs) at 48 hr (n=3; unpaired Student's t-test; **, p<0.001; ****, p<0.0001; compared to controls at the corresponding dose). (**B**) MTT assay indicates reduced GBM cell proliferation by genetic inhibition of the msiCAT-tailing pathway (n=3; unpaired Student's t-test; **, p<0.01; ***, p<0.001; ****, p<0.0001, compared to controls at the corresponding time). (**C**) MTT assay indicates reduced NHA cell proliferation by genetic inhibition of the msiCAT-tailing pathway (n=3; unpaired Student's t-test; **, p<0.01; ****, p<0.0001, compared to controls at the corresponding time). (**D, F**) Transwell assay reveals that both genetic (**D**) and pharmacological (**F**) inhibition of the msiCAT-tailing pathway hampers the migration of GBM cells but not control cells. (**E, G**) Quantification of (**D, F**) showing the number of migrated cells (n=3; unpaired Student's t-test; ***, p<0.001; ****, p<0.0001; ns, not significant). (**H, J**) TUNEL staining reveals that both genetic (**H**) and pharmacological (**J**) inhibition of the msiCAT-tailing pathway promote staurosporine (STS)-induced apoptosis in GBM cells, utilizing TUNEL-Cy3 as an apoptotic cell marker and DAPI as a nuclear stain. (**I, K**) Quantification of (**H, J**) showing the percentage of TUNEL-positive cells in the population (n=3; unpaired Student's t-test; ****, p<0.0001), using DMSO as the vehicle control. (**L**) MTT assay shows that pharmacological inhibition of the msiCAT-tailing pathway decreases the resistance of GBM cells to temozolomide (TMZ, 150 μM) treatment (n=3; unpaired Student's t-test; * p<0.05; **, p<0.01). (**M**) The neurosphere formation assay shows that reduced spheroid formation, caused by pharmacological inhibition of the msiCAT-tailing pathway, can synergize with TMZ in GBM cells. (**N**) Quantification of (**M**) (n=3; unpaired Student's t-test; **, p<0.01).

The online version of this article includes the following source data and figure supplement(s) for figure 5:

**Source data 1.** Numerical source data shown in *Figure 5A, B, C, E, G, I, K, L, and N*.

**Figure supplement 1.** No effect of cycloheximide on glioblastoma multiforme (GBM) apoptosis response.

*Figure 5 continued on next page*

*Figure 5 continued*

**Figure supplement 1—source data 1.** PDF file containing original western blots for *Figure 5—figure supplement 1B and C*, indicating the relevant bands and treatments.

**Figure supplement 1—source data 2.** Original files for western blot analysis shown in *Figure 5—figure supplement 1B and C*.

**Figure supplement 1—source data 3.** Numerical source data shown in *Figure 5—figure supplement 1A, E, and G*.

pathway's involvement in a *Drosophila* model of Notch overexpression-induced brain tumors (*Khaket et al., 2024*). The findings imply modulating the RQC pathway could serve as a promising complementary strategy to existing chemotherapy regimens. By targeting this specific pathway, therapeutic interventions might effectively disrupt the mechanisms that allow cancer cells to evade apoptosis and

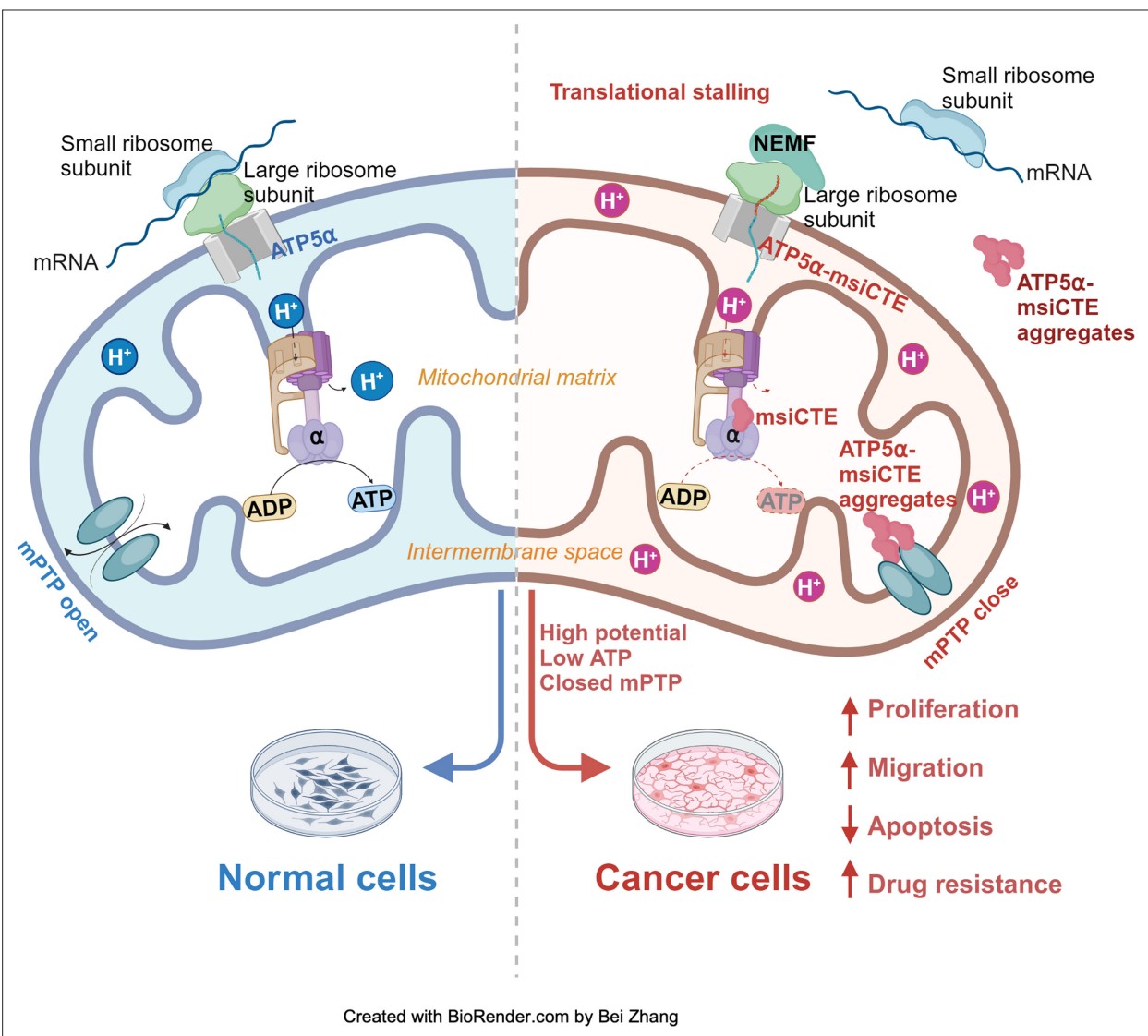

**Figure 6.** Impact of msiCAT-tail-modified ATP5α protein on mitochondrial function in glioblastoma multiforme (GBM) cells. In healthy cells, ATP5α protein, encoded by the nuclear genome, is imported into the mitochondrial matrix via the TOM/TIM complex through co-translational import and incorporated into ATP synthase (left). Conversely, in GBM cells, the CAT-tailed ATP5α protein can either form aggregates near the mitochondrial outer membrane or be imported into the mitochondria. Within the mitochondrial matrix, proteins with shorter CAT-tails readily integrate into ATP synthase, disrupting its functionality. This dysfunction is characterized by a reduced ATP synthesis rate and proton ($H^+$) accumulation, resulting in an elevated mitochondrial membrane potential ($\Delta\Psi m$). These alterations in ATP synthase ultimately trigger malfunction of the mitochondrial permeability transition pore (MPTP), consequently affecting cell proliferation, migration, and resistance to drug-induced apoptosis (right). Created with BioRender.com.

sustain their energy production under stress, potentially leading to improved treatment outcomes for patients with GBM and other cancers characterized by similar protein modifications.

The study of ATP synthase behavior in cancer holds particular importance. During carcinogenesis, ATP synthase frequently relocates to the plasma membrane, where it is referred to as ectopic ATP synthase (eATP synthase). These eATP synthases exhibit catalytic activity, facilitating ATP production in the extracellular space to foster a favorable tumor microenvironment (*Comelli et al., 2016*). Research indicates that eATP synthase assembles initially in mitochondria before being transported to the cell surface via microtubules (*Chang et al., 2023*). However, the specific type of ATP synthase delivered to the plasma membrane remains unclear. Future investigations into the localization of CAT-tailed eATP synthase may offer valuable insights into this process.

Multiple mitochondrial proteins in cancer cells can likely undergo CAT-tailing in a similar way. These msiCAT-tailed peptides may have varied impacts on mitochondria and cells due to differences in their base proteins. For instance, CAT-tailed COX4 protein might substantially and directly diminish mitochondrial respiratory efficiency. Examining the individual roles of these proteins is important, as the combined effect of their defects may be crucial in understanding observed mitochondrial changes in cancer. A minor caveat here is that the observed effect of the CAT-tails' presence primarily stems from artificial CAT-tail sequences with a high threonine content, rather than the endogenous CAT-tail protein. It is possible that other sequence components could lead to different effects (*Chang et al., 2024*). A recent study found that ANKZF1 knockdown inhibited GBM progression by causing abnormal protein accumulation in mitochondria (*Li et al., 2024*). This, combined with our data, suggests that balanced ANKZF1 expression and activity are vital for cancer proliferation. Both excess and deficiency may alter cellular adaptability. A minor flaw of that study was the use of a mitochondrial-localized nonstopped GFP protein to induce proteostasis stress and the lack of direct biochemical evidence of CAT-tailed proteins. Our research focuses on endogenous proteins for a detailed analysis of their impact on mitochondria. The rationale is that highly expressed, nonphysiological ectopic proteins might cause general proteostasis failure, masking the specific functions of endogenous proteins. Additionally, the studies used different cell lines. GSC, a patient-derived GBM cell line with greater stemness, might have a distinct mitochondrial status and RQC pathway activity compared to U87 or U251 cell lines. Thus, the conclusions of the two studies are not contradictory but rather complementary, both demonstrating the significance of RQC in tumorigenesis. Our study delves into the mechanistic role of the RQC pathway in GBM, identifying new potential targets for future treatments.

An in-depth investigation into the quantification of nuclear genome-encoded mitochondrial proteins modified via the msiCAT-tailing mechanism using sophisticated mass spectrometry is a compelling area for future research. Recent work by Lv et al., published in *Cell Reports*, revealed that the cytoplasmic E3 ligase Pirh2 and the mitochondrial protease ClpXP work in conjunction with the established NEMF-ANKZF1 system to break down mitochondrial protein aggregates resulting from ribosome stalling (*Lv et al., 2024*). The increased presence of ClpXP in various cancers could potentially be linked to an increase in msiCAT-tailing products in mitochondria, though further studies are needed to clarify ClpXP's role in mitochondrial RQC (*Cormio et al., 2021*). Moreover, ClpXP influences the levels of multiple mitochondrial proteins. Our own experiments showed that ATP5α proteins lacking msiCAT-tails were the most challenging to express ectopically. Proteins with shorter tails (AT3) expressed more readily, while those with longer tails (AT20) exhibited the highest expression levels but also tended to form SDS-insoluble aggregates. This regulatory effect could be mediated by ClpXP-dependent degradation or potentially through transcriptional control. PGC-1α, the peroxisome proliferator-activated receptor gamma co-activator, is a key regulator of mitochondrial biogenesis in mammals (*Ventura-Clapier et al., 2008*). By binding to and activating nuclear transcription factors, PGC-1α triggers the transcription of nuclear genome-encoded mitochondrial proteins and the mitochondrial transcription factor Tfam. Tfam, in turn, activates mitochondrial genome transcription and replication (*Wu et al., 1999*). Distinguishing between these regulatory possibilities will necessitate future research, including a meticulous examination of mRNA levels for msiCAT-tailed targets and analysis of PGC-1α and Tfam binding to transcriptional elements.

MPTP is a complex, supramolecular channel traversing the inner mitochondrial membrane, characterized by its nonselective ion permeability, calcium dependence, and multifaceted functionality. Despite extensive investigations into its functional attributes and regulatory mechanisms, the precise molecular architecture of the MPTP remains elusive (*Endlicher et al., 2023*). Several theoretical models

have been posited to elucidate the MPTP's structural composition. First, the VDAC/ANT/CypD model (*Beutner et al., 1998*) proposed an assembly of voltage-dependent anion channels (VDAC), adenine nucleotide translocators (ANT), and CypD as the structural basis; however, subsequent genetic analyses have introduced substantial controversy regarding the integral role of these proteins within the MPTP complex (*Baines et al., 2007*; *Gutiérrez-Aguilar et al., 2014*; *Kokoszka et al., 2004*; *Karch et al., 2019*). Second, the ATP synthase model posits that MPTP formation involves dimers or reconstituted c-rings of ATP synthase (*Alavian et al., 2014*; *Giorgio et al., 2013*). While this hypothesis presents an intriguing perspective, empirical confirmation of ATP synthase's role as a definitive structural element of the pore remains inconclusive, with a body of conflicting research surrounding this proposition. Third, the contemporary prevailing hypothesis suggests the MPTP is constituted by a large complex, termed the ATP synthasome, comprising ANT and ATP synthase, with CypD serving a regulatory function over the complex's dynamic behavior (*Beutner et al., 2017*).

The MPTP activity is modulated by mitochondrial membrane potential ($\Delta \Psi m$), which reciprocally influences mitochondrial ion homeostasis and energy metabolism (*Petronilli et al., 1994*; *Boyman et al., 2019*). Our study elucidates a dual function of msiCAT-tailed ATP5α protein in cancer cells: stabilization of a high membrane potential, thereby mitigating MPTP induction, and direct inhibition of MPTP functionality through participation in its assembly. While MPTP's critical role in cell death is established, the premise that MPTP inhibition enables cancer cell evasion of drug-induced programmed cell death has lacked substantial evidence. This study furnishes empirical support for this hypothesis, demonstrating that GBM cells, notably GSCs, exhibit markedly reduced MPTP activity relative to control cells. This reduced activity is directly correlated with the CAT-tailing modification of the ATP synthase subunit. These observations are concordant with prior research, indicating that genetic mutations or posttranslational modifications in specific ATP synthase subunits can modulate MPTP activity. The findings highlight a novel mechanism through which cancer cells may develop resistance to therapeutic interventions by manipulating mitochondrial function (*Antoniel et al., 2018*; *Carraro et al., 2020*).

## Materials and methods
### Cell lines and cell culture conditions

The human astroglia cell line SVG p12 (ATCC, cat. CRL-8621) and the human glioma cell line SF268 were from Dr. Rongze Olivia Lu. Both cell lines were cultured in DMEM (ATCC, cat. #302002) with 10% fetal bovine serum (FBS) (Biowest, cat. S1620-100) and penicillin/streptomycin (Gibco, cat. 15140122). SF268 clones should be maintained in complete DMEM supplemented with 400 µg/mL G418 (Gibco, cat. 10131027). The 0.25% trypsin solution (ATCC, cat. #SM2003C) was used to passage cells. The normal human astrocytes NHA E6/E7/hTERT cell line was from Dr. Russell O Pieper, UCSF Brain Tumor Research Center. Cells are cultured in ABM Basal Medium (Lonza, cat. CC-3187) and AGM Single-Quots Supplements (Lonza, cat. CC-4123). Corning Accutase Cell Detachment Solution (Corning, cat. 25,058CI) was used to passage cells. GSC827, a patient-derived human glioma stem cell line, was from Dr. Chun-Zhang Yang at NIH. The NSC, NSC26, patient-derived GSC33, GSC22, GSC99, GSC105, and GSC107 cell lines used in this study were kindly provided by Dr. John S Kuo at the University of Texas, Austin. Derivation of these lines from patient GBM specimens is described earlier (*Clark et al., 2012*). Detailed characterizations of the GSC lines (not GSC 105 and 107) are available in their previous publication (*Zorniak et al., 2012*). GSC 105 and 107 are not previously published. GSCs were cultured in Neural basal-A Medium (Gibco, cat. #10888022) with 2% B27 (Gibco, cat. #17504044), 1% N2 (Gibco, cat. #17502048), 20 ng/mL of EGF and FGF (Shenandoah Biotechnology Inc, cat. PB-500-017), Antibiotic-Antimycotic (Gibco, cat. #15240062), and L-Glutamine (Gibco, cat. #250300810). Cells could be cultured in both spherical and attached (on Geltrex, Thermo Fisher, cat. A1413202) forms. Corning Accutase Cell Detachment Solution (Corning, cat. 25058CI) was used to passage cells.

Cells were transfected with X-tremeGENE HP DNA Transfection Reagent (Sigma, cat. 6366244001) following the standard protocol. For single clone selection, SF268 cells were treated with 800 µg/mL G418 for 5 days. The cells were then seeded into a 96-well plate at a density of 1/100 µL. Positive clones were verified by immunofluorescence staining and immunoblotting. Cells were maintained in complete DMEM containing 400 µg/mL G418. GBM cell lines were subjected to a 4 hr pre-treatment

at 37°C using either anisomycin (20 nM or 200 nM, Fisher Scientific, cat. AAJ62964MF) or cyclo-heximide (100 µg/mL, Fisher Scientific, cat. AC357420010) in medium, as detailed in the conducted experiments.

## Primers, plasmids, and viruses

Plasmids pcDNA3.1+/C-(K)-DYK-ATP5F1A (pATP5α control), pcDNA3.1+/C-(K)-DYK-ATP5F1A-AT3 (pATP5α-AT3), pcDNA3.1+/C-(K)-DYK-ATP5F1A-AT20 (pATP5α-AT20), pcDNA3.1+/C-(K)-DYK-ATP5F1A-GS3 (pATP5α-GS3), and pcDNA3.1+/C-(K)-ATP5F1A-DYK-GS20 (pATP5α-GS20) were generated by GenScript Inc Plasmids pCMV-5×FLAG-β-globin-control (5FBG-Ctrl) and pCMV-5×FLAG-β-globin-nonstop (5FBG-nonstop) were generated by Dr. Hoshino (Nagoya City University) and Dr. Inada (Tohoku University) (*Saito et al., 2013*). pCMV6-DDK-NEMF (oeNEMF) was from ORIGENE Inc (cat. RC216806L3).

Viruses (and plasmids used to generate viruses) are pLV[CRISPR]-hCas9:T2A:Neo-U6>Scramble[gRNA#1] (sgControl/sgCtrl), pLV[CRISPR]-hCas9:T2A:Neo-U6>hNEMF[gRNA#1579] (sgNEMF), pLV[Exp]-Bsd-EF1A>ORF_Stuffer (pLV-control), pLV[Exp]-EGFP:T2A:Puro-EF1A>mCherry (pLV-control-2/oeCtrl), pLV[Exp]-Bsd-EF1A>hANKZF1[NM_001042410.2]/HA (oeANKZF1), and pLV[-Exp]-mCherry/Neo-EF1A>hANKZF1[NM_001042410.2] (oeANKZF1) were made by VectorBuilder Inc.

Primers (5' to 3') used for RT-PCR are:

| | |
|---|---|
| *LONP1* NR_076392.2 | lonp1_forward: TGCCTTGAACCCTCTCTAC |
| | lonp1_reverse: TCTGCTTGATCTTCTCCTCC |
| *mtHSP70* NM_004134.7 | mthsp70_forward: ACTCCTCCATTTATCCGCC |
| | mthsp70_reverse: ACCTTTGCTTGTTTACCTTCC |
| *HSP60* NM_002156.5 | hsp60_forward: ACCTGCTCTTGAAATTGCC |
| | hsp60_reverse: CAATCCCTCTTCTCCAAACAC |
| *ACTB* NM_001101.5 | actb_forward: TGTTTGAGACCTTCAACACC |
| | actb_reverse: ATGTCACGCACGATTTCC |

## Neurosphere formation assay of GSCs

The GSC spheroids were dissociated using Accutase for 2 min. Cells were resuspended in a single-cell suspension and grown under nonadherent conditions. Cells were seeded in 12-well plates at a density of $0.25 \times 10^6$ cells/well and cultured in 3 mL culture medium for 24 hr. 20 nM of anisomycin and 150 µM of TMZ were added to the culture medium, and the cells were treated for 96 hr. Spheroids were imaged under a 10× objective, captured using QCapture, and analyzed with ImageJ. Spheroids larger than 50 µm were counted.

## Differential gene expression analysis using the public database

The raw RNA-seq data used for the analysis were obtained from the University of California, Santa Cruz Xenabrowser (cohort: TCGA TARGET GTEx, dataset ID: TcgaTargetGtex_rsem_gene_tpm, https://xena.ucsc.edu/). Subsets were then created to include only TCGA glioma (GBM), GTEx Brain Frontal Cortex, and GTEx Cortex samples. Differential expression analysis was conducted using the 'Limma' package (R version: 4.3.1). The Benjamini-Hochberg method was used for multiple testing correction to control the false discovery rate. Cutoff of adjusted p-value (adj.P.Val) was set at 0.001; cutoff of the absolute fold change was set at 2 (logFC>1). The code is available without restrictions at https://github.com/yuanna23/GBM_elife, (copy archived at *Wu, 2026*).

## Immunostaining

Cells were cultured on sterile coverslips until 80% confluency. For immunostaining, cells were washed with phosphate-buffered saline (PBS) solution thrice. Then, 4% formaldehyde (Thermo Fisher, cat. BP531-500) was applied to cells for fixation for 30 min at room temperature. After fixation, cells were washed with PBS solution containing 0.25% Triton X-100 (PBSTx) (Thermo Fisher, cat. T9284) thrice and blocked with 5% normal goat serum (Jackson Immuno, cat. 005-000-121) for 1 hr at room

temperature. Cells were then incubated with primary antibodies overnight in a humidified chamber at 4°C. The next day, cells were washed by PBSTx thrice and incubated with secondary antibodies for 2 hr at room temperature. After washing, cells were stained with 300 nM DAPI (Thermo Fisher, cat. 57-481-0) for 5 min at room temperature and mounted in Fluoromount-G Anti-Fade solution (Southern Biotech, cat. 0100-35). Images were captured using a Zeiss LSM 800 confocal microscope with a 40× oil objective lens and AiryScan processing. The primary antibodies used in the study were rabbit anti-ATP5a (Cell Signaling, cat. #18023), mouse anti-TOMM20 (1:500, Santa Cruz, cat. sc-17764), rabbit anti-MTCO2 (1:500, Proteintech, cat. 55070-1-AP), and mouse anti-NDUS3 (1:1000, Abcam, cat. ab14711). The secondary antibodies were Alexa Fluor 633-, 594-, 488-conjugated secondary antibodies (1:300, Invitrogen, cat. A21071, A11036, A32732).

## SDS-PAGE and immunoblotting

Cells or isolated mitochondria were solubilized in cell lysis buffer containing 50 mM Tris-HCl pH 7.4, 150 mM NaCl, 10% glycerol, 1% Triton X-100, 5 mM EDTA, and 1× protease inhibitor (Bimake, cat. B14002). Protein concentration was measured by using the Bradford assay (BioVision, cat. K813-5000-1). Samples were separated in a 4–12% Tris-Glycine gel (Invitrogen, cat. WXP41220BOX), and proteins were transferred to a PVDF membrane (Millipore, cat. ISEQ00010). The membranes were then blocked with 5% nonfat dry milk (Kroger) for 50 min at room temperature and probed with primary antibodies overnight at 4°C. Membranes were washed with Tris-buffered saline with 0.1% Tween 20 (TBST) solution thrice and then incubated with secondary antibodies for 1 hr at room temperature. Blots were detected with ECL solution (PerkinElmer, cat. NEL122001EA) and imaged by Chemidoc system (Bio-Rad). The intensity of blots was further analyzed by ImageJ software. The primary antibodies used were mouse anti-Actin (1:1000, Santa Cruz, cat. sc-47778), rabbit anti-NEMF (1:1000, Proteintech, cat. 11840-1-AP), mouse anti-ANKZF1 (1:1000, Santa Cruz, cat. sc-398713), mouse anti-ATP5a (Abcam, cat. ab14748), mouse anti-NDUS3 (1:1000, Abcam, cat. ab14711), rabbit anti-COX4 (Abcam, cat. ab209727), mouse anti-Flag (1:1000, Sigma, cat. F1804), rabbit anti-ANT1/2 (1:1000, Proteintech, cat. 17796-1-AP), rabbit anti-CypD (1:1000, Proteintech, cat. 15997-1-AP), rabbit anti-PARP1 (1:1000, Abclonal, cat. A0942), rabbit anti-GAPDH (1:1000, Abclonal, cat. A19056). The secondary antibodies used were goat anti-rabbit IgG (1:5000, Invitrogen, cat. G21234) and goat anti-mouse IgG (1:5000, Invitrogen, cat. PI31430).

## Mitochondrial isolation, BN-PAGE, and western blotting

Cells were homogenized using a Dounce homogenizer in ice-cold homogenization buffer containing 210 mM mannitol (Fisher Science, cat. AA3334236), 70 mM sucrose (Fisher Science, cat. AA36508A1), 5 mM HEPES (Fisher Science, cat. 15630106), pH 7.12, 1 mM EGTA (Fisher Science, cat. 28-071-G), and 1× protease inhibitor. The homogenate was centrifuged at 1500×$g$ for 5 min. The resultant supernatant was centrifuged at 13,000×$g$ for 17 min. The supernatant was collected as the cytosol portion, and the pellet (the mitochondria portion) was washed with homogenization buffer and centrifuged at 13,000×$g$ for 10 min. For BN-PAGE, the mitochondrial samples were solubilized in 5% digitonin (Thermo Fisher, cat. BN2006) on ice for 30 min and then centrifuged at 20,000×$g$ for 30 min. The supernatant contains solubilized mitochondrial proteins and was mixed with 5% G-250 (GoldBio, cat. C-460-5) and 1× NativePAGE sample buffer (Invitrogen, cat. BN2008) (final G-250 concentration is 25% of the digitonin concentration). Mitochondrial protein concentration was measured by using the Bradford assay. Samples were separated in 3–12% Bis-Tris Native gel (Invitrogen, cat. BN1001BOX) and then transferred to a PVDF membrane. Membranes were fixed with 8% acetic acid (Thermo Fisher, cat. 9526-33), and then blocked and probed with antibodies as described above for western blotting.

## Mitochondrial membrane potential assays

Mitochondrial membrane potential of GSCs was measured using Image-iT TMRM (Invitrogen, cat. I34361). Cells were cultured in 96-well black plates at a density of 1×10⁵ cells per well overnight in an incubator with 5% $CO_2$ at 37°C. Cells were incubated with TMRM (100 nM) for 30 min at 37°C. Then, cells were washed with PBS solution three times. Fluorescence changes at excitation/emission of 548/574 nm were monitored with a Cytation 5 plate reader (BioTek). Mitochondrial membrane potential was also measured using JC-10 (AdipoGen, cat. 50-114-6552). Cells were cultured in 96-well black plates at a density of 5×10⁴ cells per well overnight in an incubator with 5% $CO_2$ at 37°C. Cells were

incubated with JC-10 (10 µg/mL) for 45 min at 37°C. Then, cells were washed with PBS solution twice. Fluorescence changes at excitation/emission of 535/595 nm for JC-10 aggregates and at 485/535 nm for JC-10 monomers were monitored with a Synergy 2 Reader (BioTek). Mitochondrial membrane potential was quantified as the fluorescence of JC-10 aggregates/monomers (595/535 nm).

## Seahorse cell mitochondrial stress assays

The OCR of cells was measured using the Seahorse Cell Mito Stress Test kit following the user guide (Agilent, cat. 103010-100). Briefly, cells were cultured overnight in testing chambers at a density of 8000 cells per well in an incubator with 5% $CO_2$ at 37°C. Cells were then washed twice with assay medium containing Seahorse XF DMEM medium (Agilent, cat. 103575-100) supplemented with 1 mM pyruvate, 2 mM glutamine, and 10 mM glucose. They were subsequently incubated in the assay medium for 1 hr in an incubator without $CO_2$ at 37°C. Cells were treated with compounds in the order of oligomycin (1.5 µM), carbonyl cyanide-4 (trifluoromethoxy), phenylhydrazone (FCCP, 1.0 µM), and rotenone/antimycin (0.5 µM). The OCR of cells was monitored by using Seahorse XF HS Mini (Agilent).

## Mitochondrial MPTP assay

The status of MPTP was measured using Invitrogen Image-IT LIVE Mitochondrial Transition Pore Assay Kit (Invitrogen, cat. I35103). Cells were cultured in 35 mm glass-bottom dishes overnight in an incubator with 5% $CO_2$ at 37°C. Cells were washed twice with the modified Hank's Balanced Salt Solution (HBSS, Thermo Fisher, cat. 14025092) containing 10 mM HEPES, 2 mM L-glutamine, and 0.1 mM succinate (Thermo Fisher, cat. 041983.A7) and incubated with the labeling solution (1 µM Calcein, 0.2 µM MitoTracker Red, 1 mM Cobalt Chloride) for 15 min at 37°C. Cells were then washed with HBSS twice and imaged at excitation/emission of 494/517 nm for Calcein and at 579/599 nm for MitoTracker Red by using the Zeiss confocal microscope.

## Mitochondrial CRC assay

The mitochondrial CRC was measured on a Cytation 5 reader at excitation/emission of 506/592 nm using the membrane-impermeable fluorescent probe Calcium Green-5N (Invitrogen, cat. C3737). Isolated mitochondria samples (0.75 mg protein/mL) were incubated in 1 mL swelling medium supplemented with 10 mM succinate, 1 µM Calcium Green-5N, inorganic phosphate, and cyclosporine A (Thermo Fisher, cat. AC457970010). One $Ca^{2+}$ addition was 1.25 nmol (1 mL volume). Only the MPTP opening in the presence of cyclosporine A was induced by high amounts of added calcium (30 nmol $Ca^{2+}$ in the last two additions). The CRC value was calculated as total $Ca^{2+}$ accumulated in the mitochondria per unit (1 mg protein).

## MTT assay

Cell proliferation was measured by using the MTT assay kit (Roche, cat. 11465007001). Cells were cultured in 96-well plates at a density of 2000 cells per well overnight in an incubator with 5% $CO_2$ at 37°C. Cells were treated with MTT labeling reagent for 4 hr at 37°C. The solubilization buffer was added to the cells, and then the cells were incubated overnight at 37°C. Absorbance changes of the samples at 550 nm were monitored by using a Synergy 2 Reader (BioTek).

## Wound healing assay

Cells were seeded into six-well plates and cultured for 24–48 hr to reach a confluent cell monolayer. Cells were treated with serum-free medium overnight before mechanical scratching (*Grada et al., 2017*). Images of the wounds were taken at 0, 24, and 48 hr. Wound areas were measured by using the wound healing plugin of ImageJ. Wound coverage %=100% x $[A_t - A_{t=Dh}]/A_t$ ($A_t$ is the area of the wound measured immediately after scratching t=0 hr, $A_{t\,Dh}$ is the area of the wound measured h hours after the scratch is performed).

## Cell migration assay

Cell migration was measured by using Transwell assays (Corning, cat. CLS3422). Cells were cultured in Transwell inserts at a density of $1\times10^5$ cells per well for 3 hr in an incubator at 37°C with 5% $CO_2$. The top inserts were supplemented with DMEM medium only, and the bottom wells were supplemented with DMEM medium with 20% FBS. After incubation, the cells on the apical side of the Transwell

insert membrane were removed using a cotton applicator. The cells on the bottom side of the insert were rinsed with PBS twice and fixed in 70% ethanol (Thermo Fisher, cat. R40135) for 15 min at room temperature. After fixation, inserts were placed into an empty well to allow the membrane to dry. Then, the insert was incubated with 0.2% crystal violet (Sigma, cat. V5265) for 5 min at room temperature. The insert was rinsed with water twice, and images were captured by using a microscope with a 20× objective. Cell numbers were quantified using ImageJ.

## TUNEL staining

The apoptosis was measured by a TUNEL assay kit (ApexBio, cat. K1134). Cells were cultured on sterile coverslips until 80% confluency and washed with PBS thrice. Then, 4% formaldehyde was applied to cells and fixed at 4°C for 25 min. After fixation, cells were washed with PBS twice and incubated with 20 μM proteinase K (Invitrogen, cat. 25530049) for 5 min at room temperature. Then, cells were rinsed with PBS thrice and incubated in 1× equilibration buffer for 10 min at room temperature. Cells were stained with FITC or Cy3 labeling mix for 1 hr at 37°C in a humidified chamber. Cells were washed with PBS thrice and stained with DAPI for 5 min at room temperature. Cells were mounted in the Fluoromount-G Anti-Fade solution and imaged at 520 nm for FITC or at 570 nm for Cy3 by using the Zeiss confocal microscope.

## Caspase-3/7 activity assay

Caspase-3/7 activity was measured by using CellEvent Caspase-3/7 Detection Reagents (Invitrogen, cat. C10432) following the manufacturer's protocol. Specifically, cells were seeded in a 96-well black plate with a clear bottom at a density of $5 \times 10^4$ cells per well and incubated overnight in the incubator with 5% $CO_2$ at 37°C. Cells were then incubated with 1× staining solution for 30 min at 37°C. Fluorescence changes at excitation/emission of 485/525 nm were monitored with a Synergy 2 Reader (BioTek).

## Annexin V-FITC/PI apoptosis detection

Annexin V-FITC/Propidium Iodide (PI) apoptosis assay was performed by using the FITC Annexin V Apoptosis Detection Kit with PI (BioLegend, cat. 640914). Briefly, $1 \times 10^5$ cells were collected in 100 μL of staining buffer. Then, cells were incubated with 5 μL of Annexin V-FITC and 2.5 μL of PI for 15 min at room temperature in the dark. Following incubation, 400 μL of binding buffer was added to the stained cells. Flow cytometry analysis of the fluorescence was performed using a Soni SH800 Cell Sorter.

## Mitochondria ATP measurement via fluorescence imaging of ATP-red

BioTracker ATP-red dye (Millipore, cat. SCT045) is a fluorogenic indicator for ATP in mitochondria (*Wang et al., 2016*). Cells cultured in monolayer conditions were incubated in medium with 5 μM ATP-red for 15 min in an incubator at 37°C with 5% $CO_2$. Mitochondria were also labeled by incubating cells with 100 nM MitoTracker-Green (Invitrogen, cat. M7514) for 15 min to normalize their mass. Before measurement, cells were washed twice with culture medium, and then fresh medium was added. Cells were imaged in a 37°C chamber with 5% $CO_2$ at excitation/emission of 510/570 nm for ATP-red and at excitation/emission of 490/516 nm for MitoTracker-Green by using the Zeiss confocal microscope. The ATP-red signals could also be measured by a Synergy 2 Reader (BioTek).

## Co-immunoprecipitation

Cells were lysed in the buffer containing 50 mM Tris-HCl pH 7.4, 150 mM NaCl, 10% glycerol, 1% Triton X-100, 5 mM EDTA, and 1× protease inhibitor. Soluble samples were incubated with 1.5 μL ATP5α antibody at 4°C with mixing overnight. 25 μL of protein A/G magnetic beads (Pierce, cat. 88802) were added to the co-IP samples and incubated at 4°C with mixing overnight. Samples were washed with washing buffer thrice and then applied to SDS-PAGE analysis.

## Mice and immunostaining

Animal studies were approved by the University of California, San Francisco Institutional Animal Care and Use Committee (IACUC, AN195636-01) and were performed following the guidelines of the National Institutes of Health (NIH).

For orthotopic brain tumor models, 8- to 10-week-old C57BL/6J mice (male and female in equal numbers) were used for i.c. studies. Cell lines (GL261, SB28) were suspended in DMEM for inoculation. Mice were anesthetized with isoflurane, and 30,000 tumor cells were injected orthotopically in 3 µL. Using a stereotactic frame, a burr hole was formed on the skull via a 0.7 mm drill bit 1.5 mm laterally to the right and 1.5 mm rostrally from the bregma, and a noncoring needle (26s gauge; Hamilton) was used to inject the cells at a depth of 3 mm into the brain from the burr hole. The skin incision was sutured. Mice were then monitored daily. Mouse SB28 tumor tissue and wild-type mouse brain tissue were collected at the survival endpoint.

Frozen tissue sections were thawed at room temperature for 20 min and rinsed with PBS three times. Tissues were then fixed in 4% formaldehyde for 15 min at room temperature. After washing in PBS, tissues were permeabilized with 0.01% Triton X-100+0.1% Tween-20 for 15 min and then blocked by using 5% normal goat serum and M.O.M. blocking reagent (Vector Laboratories, cat. BMK-2202) for 1 hr at room temperature. Tissues were then incubated with primary antibodies overnight in a humidified chamber at 4°C. After washing in PBST, tissues were incubated with secondary antibodies for 1 hr. After washing again in PBST, tissues were stained with 300 nM DAPI for 5 min and mounted in Fluoromount-G Anti-Fade solution. Images were taken using a Zeiss LSM 800 confocal microscope. The primary antibodies used in the study were mouse anti-ATP5a (1:500, Abcam, cat. Ab14748), rat anti-TOMM20 (1:500, Abcam, cat. Ab289670), rabbit anti-NEMF (1:500, Proteintech, cat. 11840-1-AP), mouse anti-ANKZF1 (1:500, Santa Cruz, cat. sc-398713), and chicken anti-GFP (1:500, Abcam, cat. Ab13970). The secondary antibodies were Alexa Fluor 633-, 594-, 488-conjugated secondary antibodies (1:300, Invitrogen, cat. A21071, A11036, A32732).

## Statistics

Statistical analyses were performed using GraphPad Prism 9.4. Chi-squared test and unpaired Student's t-test were used for comparison. $p < 0.05$ was considered significant, except in gene expression analysis (*Figure 1A*). *, $p < 0.05$; **, $p < 0.01$; ***, $p < 0.001$; ****, $p < 0.0001$; ns, not significant. All data were expressed as means ± s.e.m. This study's replicates, samples, groups, and experiments were biologically independent, except in *Table 1*. The 'n' numbers for each assay are indicated in the figure legends.

## Materials availability

Plasmids and other reagents generated in this study will be made available to researchers by contacting zhihaowu@smu.edu. The patient-derived materials are proprietary to Prof. John S Kuo, but available on personal requests via standard institution/university agreements.

## Code availability

The code used for differential gene expression analysis is available without restrictions at https://github.com/yuanna23/GBM_elife, (*Wu, 2026*).

## Acknowledgements

The patient-derived GSC and NSC lines were a gift from Dr. John S Kuo at the University of Texas, Austin. We are also grateful to Dr. Chunzhang Yang at the National Cancer Institute and Dr. Russell O Pieper at the University of California, San Francisco, for the cell lines. We thank Dr. Hoshino at Nagoya City University and Dr. Inada at Tohoku University for providing the plasmids. We also thank the members of the Wu lab at Southern Methodist University and the Lu lab at UCSF for their assistance and discussions. National Institute of General Medical Sciences (R35GM150190), Zhihao Wu. National Institute of Neurological Disorders and Stroke (R01NS126501), Rongze Olivia Lu. Cancer Prevention and Research Institute of Texas (RP210068), Zhihao Wu and Rongze Olivia Lu. Southern Methodist University (Dedman College Dean's Research Council grant), Zhihao Wu. SPARK-South Carolina Alzheimer's Disease Research Center (ARDC Pilot Grant), Qing Liu. The funders had no role in the study design, data collection, and interpretation, or in the decision to submit the work for publication.

# Additional information

## Funding

| Funder | Grant reference number | Author |
|---|---|---|
| National Institute of General Medical Sciences | R35GM150190 | Zhihao Wu |
| National Institute of Neurological Disorders and Stroke | R01NS126501 | Rongze Olivia Lu |
| Cancer Prevention and Research Institute of Texas | RP210068 | Rongze Olivia Lu Zhihao Wu |
| Southern Methodist University | Dedman College Dean's Research Council grant | Zhihao Wu |
| South Carolina Alzheimer's Disease Research Center | ARDC Pilot Grant | Qing Liu |

The funders had no role in study design, data collection and interpretation, or the decision to submit the work for publication.

## Author contributions

Bei Zhang, Data curation, Formal analysis, Validation, Investigation, Visualization, Methodology, Writing – original draft; Ting Cai, Data curation, Formal analysis, Investigation, Visualization, Methodology, Writing – original draft; Esha Reddy, Isha Mondal, Yinglu Tang, Adaeze Scholastical Gbufor, Jerry Wang, Yawei Shen, Raymond Sun, Investigation, Writing – review and editing; Yuanna Wu, Software, Formal analysis, Writing – review and editing; Qing Liu, Funding acquisition, Investigation, Writing – review and editing; Winson S Ho, Resources, Investigation, Writing – review and editing; Rongze Olivia Lu, Conceptualization, Resources, Supervision, Funding acquisition, Investigation, Project administration, Writing – review and editing; Zhihao Wu, Conceptualization, Resources, Supervision, Funding acquisition, Investigation, Writing – original draft, Project administration, Writing – review and editing

## Author ORCIDs

Raymond Sun ⓘ https://orcid.org/0000-0001-6352-0642
Rongze Olivia Lu ⓘ https://orcid.org/0000-0003-1600-9299
Zhihao Wu ⓘ https://orcid.org/0000-0003-3080-5769

Reviewer #1 (Public review): https://doi.org/10.7554/eLife.99438.4.sa1
Reviewer #2 (Public Review): https://doi.org/10.7554/eLife.99438.4.sa2
Author response https://doi.org/10.7554/eLife.99438.4.sa3

# Additional files

## Supplementary files

MDAR checklist

## Data availability

The numerical data used to generate the figures are provided as source data.

The following previously published dataset was used:

| Author(s) | Year | Dataset title | Dataset URL | Database and Identifier |
|---|---|---|---|---|
| UCSC TOIL RNA-seq recompute | 2016 | dataset: gene expression RNAseq - RSEM tpm | https://xenabrowser.net/datapages/?dataset=TcgaTargetGtex_rsem_gene_tpm&host=https%3A%2F%2Ftoil.xenahubs.net&removeHub=http%3A%2F%2F127.0.0.1%3A7222 | UCSC Xena, TcgaTargetGtex_rsem_gene_tpm |

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

# Appendix 1

**Appendix 1—key resources table**

| Reagent type (species) or resource | Designation | Source or reference | Identifiers | Additional information |
|---|---|---|---|---|
| Strain, strain background (mice) | *C57BL/6J* mice | Dr. Rongze Olivia Lu | RRID:IMSR_JAX:000664 | |
| Cell line (*Homo sapiens*) | SVG p12 | ATCC | RRID:CRL-8621 | |
| Cell line (*H. sapiens*) | SF268 | Dr. Rongze Olivia Lu | RRID:CVCL_1689 | |
| Cell line (*H. sapiens*) | GL261 | Dr. Rongze Olivia Lu | RRID:CVCL_Y003 | |
| Cell line (*H. sapiens*) | SB28 | Dr. Rongze Olivia Lu | RRID:CVCL_A5ED | |
| Cell line (*H. sapiens*) | GSC827 | Dr. Chun-Zhang Yang | | Patient-derived |
| Cell line (*H. sapiens*) | NSC | Dr. John S Kuo | | Human-derived |
| Cell line (*H. sapiens*) | NSC26 | Dr. John S Kuo | | Human-derived |
| Cell line (*H. sapiens*) | GSC33 | Dr. John S Kuo | | Patient-derived |
| Cell line (*H. sapiens*) | GSC22 | Dr. John S Kuo | | Patient-derived |
| Cell line (*H. sapiens*) | GSC99 | Dr. John S Kuo | | Patient-derived |
| Cell line (*H. sapiens*) | GSC105 | Dr. John S Kuo | | Patient-derived |
| Cell line (*H. sapiens*) | GSC107 | Dr. John S Kuo | | Patient-derived |
| Cell line (*H. sapiens*) | NHA | Dr. Russell O Pieper | RRID:CVCL_E3G5 | |
| Transfected construct (human) | pLV[CRISPR]-hCas9:T2A:Neo-U6>Scramble [gRNA#1] | Made by VectorBuilder | Cat#: VB240227-1635qjy | Lentiviral construct to express control sgRNA. |
| Transfected construct (human) | pLV[CRISPR]-hCas9:T2A:Neo-U6>hNEMF [gRNA#1579] | Made by VectorBuilder | Cat#: VB900124-2190daq | Lentiviral construct to express human NEMF sgRNA |
| Transfected construct (human) | pLV[Exp]-Bsd-EF1A>ORF_Stuffer | Made by VectorBuilder | Cat#: VB900145-3633yjp | The control lentiviral construct to express target gene |
| Transfected construct (human) | pLV[Exp]-EGFP:T2A:Puro-EF1A>mCherry | Made by VectorBuilder | Cat#: VB010000-9298rtf | The control lentiviral construct to express target gene |
| Transfected construct (human) | pLV[Exp]-Bsd-EF1A>hANKZF1 [NM_001042410.2]/HA | Made by VectorBuilder | Cat#: VB240227-1626epe | Lentiviral construct to express human ANKZF1 gene |
| Transfected construct (human) | pLV[Exp]-mCherry/Neo-EF1A>hANKZF1 [NM_001042410.2] | Made by VectorBuilder | Cat#: VB900124-2193gcv | Lentiviral construct to express human ANKZF1 gene |

*Appendix 1 Continued on next page*

*Appendix 1 Continued*

| Reagent type (species) or resource | Designation | Source or reference | Identifiers | Additional information |
|---|---|---|---|---|
| Antibody | Anti-COX4 (Rabbit polyclonal) | Abcam | Cat#: ab209727, RRID:AB_3717302 | WB (1:1000) |
| Antibody | Anti-β-Actin [C4] (Mouse monoclonal) | Santa Cruz | Cat#: sc-47778, RRID:AB_626632 | WB (1:1000) |
| Antibody | Anti-Flag (Mouse monoclonal) | Millipore Sigma | Cat#: F1804, RRID:AB_262044 | WB (1:1000) |
| Antibody | Anti-ANT1/2 (Rabbit polyclonal) | Proteintech | Cat#: 17796-1-AP, RRID:AB_2190358 | WB (1:1000) |
| Antibody | Anti-CypD (Rabbit polyclonal) | Proteintech | Cat#: 15997-1-AP, RRID:AB_2190199 | WB (1:1000) |
| Antibody | Anti-ATP5a (Rabbit polyclonal) | Cell Signaling Technology | Cat#: 18023, RRID:AB_2687556 | WB (1:1000) IF (1:500) |
| Antibody | Anti-PARP1 (Rabbit polyclonal) | Abclonal | Cat#: A0942, RRID:AB_2757470 | WB (1:1000) |
| Antibody | Anti-GAPDH (Rabbit polyclonal) | Abclonal | Cat#: A19056, RRID:AB_2862549 | WB (1:1000) |
| Antibody | Anti-TOMM20 (Mouse monoclonal) | Santa Cruz | Cat#: sc-17764, RRID:AB_628381 | WB (1:1000) IF (1:500) |
| Antibody | Anti-MTCO2 (Rabbit polyclonal) | Proteintech | Cat#: 55070-1-AP, RRID:AB_10859832 | WB (1:1000) IF (1:500) |
| Antibody | Anti-NDUS3 (Mouse monoclonal) | Abcam | Cat#: ab14711, RRID:AB_301429 | WB (1:1000) IF (1:1000) |
| Antibody | Anti-NEMF (Rabbit polyclonal) | Proteintech | Cat#: 11840-1-AP, RRID:AB_2183413 | WB (1:1000) IF (1:500) |
| Antibody | Anti-ANKZF1 (Mouse monoclonal) | Santa Cruz | Cat#: sc-398713, RRID:AB_3094545 | WB (1:1000) IF (1:500) |
| Antibody | Anti-ATP5a (Mouse monoclonal) | Abcam | Cat#: ab14748, RRID:AB_301447 | WB (1:1000) IF (1:500) |
| Antibody | Anti-TOMM20 (Rat monoclonal) | Abcam | Cat#: Ab289670, RRID:AB_3097753 | WB (1:1000) IF (1:500) |
| Antibody | Anti-GFP (Chicken polyclonal) | Abcam | Cat#: Ab13970, RRID:AB_300798 | WB (1:1000) IF (1:500) |
| Antibody | Anti-rabbit HRP (Goat polyclonal) | Invitrogen | Cat#: G21234, RRID:AB_2536530 | WB (1:5000) |
| Antibody | Anti-mouse HRP (Goat polyclonal) | Invitrogen | Cat#: PI31430, RRID:AB_228307 | WB (1:5000) |
| Antibody | Anti-Mouse IgG (H+L) Highly Cross-Adsorbed Secondary Antibody, Alexa Fluor 488 (Goat polyclonal) | Invitrogen | Cat#: A32723, RRID:AB_2633275 | IF (1:300) |
| Antibody | Anti-Rabbit IgG (H+L) Cross-Adsorbed Secondary Antibody, Alexa Fluor 633 (Goat polyclonal) | Invitrogen | Cat#: A21071, RRID:AB_2535732 | IF (1:300) |

*Appendix 1 Continued on next page*

*Appendix 1 Continued*

| Reagent type (species) or resource | Designation | Source or reference | Identifiers | Additional information |
|---|---|---|---|---|
| Antibody | Anti-Rabbit IgG (H+L) Highly Cross-Adsorbed Secondary Antibody, Alexa Fluor 568 (Goat polyclonal) | Invitrogen | Cat#: A11036, RRID:AB_10563566 | IF (1:300) |
| Recombinant DNA reagent | pcDNA3.1+/C-(K)-DYK-ATP5F1A | Made by GenScript | Clone ID: OHu25769D | Construct to express human ATP5F1A. Available from Wu Lab |
| Recombinant DNA reagent | pcDNA3.1+/C-(K)-DYK-ATP5F1A-AT3 | Made by GenScript | Modified from OHu25769D | Construct to express human ATP5F1A with AT3 tail. Available from Wu Lab |
| Recombinant DNA reagent | pcDNA3.1+/C-(K)-DYK-ATP5F1A-AT20 | Made by GenScript | Modified from OHu25769D | Construct to express human ATP5F1A with AT20 tail. Available from Wu Lab |
| Recombinant DNA reagent | pcDNA3.1+/C-(K)-DYK-ATP5F1A-GS3 | Made by GenScript | Modified from OHu25769D | Construct to express human ATP5F1A with GS3 tail. Available from Wu Lab |
| Recombinant DNA reagent | pcDNA3.1+/C-(K)-ATP5F1A-DYK-GS20 | Made by GenScript | Modified from OHu25769D | Construct to express human ATP5F1A with GS20 tail. Available from Wu Lab |
| Recombinant DNA reagent | pCMV-5×FLAG-β-globin-control | Dr. Hoshino and Dr. Inada | | Available from Inada Lab |
| Recombinant DNA reagent | pCMV-5×FLAG-β-globin-non-stop | Dr. Hoshino and Dr. Inada | | Available from Inada Lab |
| Recombinant DNA reagent | pCMV6-DDK-NEMF (NM_004713) | ORIGENE | Cat#: RC216806L3 | |
| Sequence-based reagent | lonp1_F | Made by GeneWiz | PCR primers | TGCCTTGAACCCTCTCTAC |
| Sequence-based reagent | lonp1_R | Made by GeneWiz | PCR primers | TCTGCTTGATCTTCTCCTCC |
| Sequence-based reagent | mthsp70_F | Made by GeneWiz | PCR primers | ACTCCTCCATTTATCCGCC |
| Sequence-based reagent | mthsp70_R | Made by GeneWiz | PCR primers | ACCTTTGCTTGTTTACCTTCC |
| Sequence-based reagent | hsp60_F | Made by GeneWiz | PCR primers | ACCTGCTCTTGAAATTGCC |
| Sequence-based reagent | hsp60_R | Made by GeneWiz | PCR primers | CAATCCCTCTTCTCCAAACAC |
| Sequence-based reagent | actb_F | Made by GeneWiz | PCR primers | TGTTTGAGACCTTCAACACC |
| Sequence-based reagent | actb_R | Made by GeneWiz | PCR primers | ATGTCACGCACGATTTCC |
| Commercial assay or kit | AGM SingleQuots Supplements | Lonza | Cat#: CC-4123 | |
| Commercial assay or kit | MTT assay kit | Roche | Cat#: 11465007001 | |
| Commercial assay or kit | Seahorse Cell Mito Stress Test kit | Agilent | Cat#: 103010-100 | |
| Commercial assay or kit | NativePAGE Running Buffer Kit | Invitrogen | Cat#: BN2007 | |

*Appendix 1 Continued on next page*

*Appendix 1 Continued*

| Reagent type (species) or resource | Designation | Source or reference | Identifiers | Additional information |
|---|---|---|---|---|
| Commercial assay or kit | NativePAGE Sample Prep Kit | Invitrogen | Cat#: BN2008 | |
| Commercial assay or kit | TUNEL assay | ApexBio | Cat#: K1134 | |
| Commercial assay or kit | Annexin V-FITC/PI apoptosis assay | BioLegend | Cat#: 640914 | |
| Commercial assay or kit | Seahorse XF DMEM medium | Agilent | Cat#: 103575-100 | |
| Commercial assay or kit | Mitochondrial Transition Pore Assay | Invitrogen | Cat#: I35103 | |
| Chemical compound, drug | DMEM | ATCC | Cat#: 302002 | |
| Chemical compound, drug | FBS | Biowest | Cat#: S1620-100 | |
| Chemical compound, drug | Penicillin/streptomycin | Gibco | Cat#: 15140122 | |
| Chemical compound, drug | G418 | Gibco | Cat#: 10131027 | |
| Chemical compound, drug | 0.25% trypsin solution | ATCC | Cat#: SM2003C | |
| Chemical compound, drug | ABM Basal Medium | Lonza | Cat#: CC-3187 | |
| Chemical compound, drug | Accutase | Corning | Cat#: 25058CI | |
| Chemical compound, drug | Neural basal-A Medium | Gibco | Cat#: 10888022 | |
| Chemical compound, drug | B27 | Gibco | Cat#: 17504044 | |
| Chemical compound, drug | N2 | Gibco | Cat#: 17502048 | |
| Chemical compound, drug | EGF and FGF | Shenandoah Biotech | Cat#: PB-500-017 | |
| Chemical compound, drug | Antibiotic-Antimycotic | Gibco | Cat#: 15240062 | |
| Chemical compound, drug | L-Glutamine | Gibco | Cat#: 250300810 | |
| Chemical compound, drug | Geltrex | Thermo Fisher | Cat#: A1413202 | |
| Chemical compound, drug | X-tremeGENE | Sigma | Cat#: 6366244001 | |
| Chemical compound, drug | Anisomycin | Fisher Scientific | Cat#: AAJ62964MF | |
| Chemical compound, drug | Cycloheximide | Fisher Scientific | Cat#: AC357420010 | |
| Chemical compound, drug | Temozolomide | Millipore Sigma | Cat#: 50-060-90001 | |
| Chemical compound, drug | Formaldehyde | Thermo Fisher | Cat#: BP531-500 | |

*Appendix 1 Continued*

| Reagent type (species) or resource | Designation | Source or reference | Identifiers | Additional information |
|---|---|---|---|---|
| Chemical compound, drug | Triton X-100 | Thermo Fisher | Cat#: T9284 | |
| Chemical compound, drug | Lipofectamine 3000 | Invitrogen | Cat#: L3000015 | |
| Chemical compound, drug | Normal goat serum | Jackson Immuno | Cat#: 005-000-121 | |
| Chemical compound, drug | DAPI | Thermo Fisher | Cat#: 57-481-0 | |
| Chemical compound, drug | Fluoromount-G Anti-Fade | Southern Biotech | Cat#: 0100-35 | |
| Chemical compound, drug | Puromycin | ARCOS organics | Cat#: 227420100 | |
| Chemical compound, drug | Protease inhibitor | Bimake | Cat#: B14002 | |
| Chemical compound, drug | Bradford | BioVision | Cat#: K813-5000-1 | |
| Chemical compound, drug | Mannitol | Fisher Scientific | Cat#: AA3334236 | |
| Chemical compound, drug | Sucrose | Fisher Scientific | Cat#: AA36508A1 | |
| Chemical compound, drug | HEPES | Fisher Scientific | Cat#: 15630106 | |
| Chemical compound, drug | Western Lightning Plus-ECL | PerkinElmer Inc | Cat#: NEL104001EA | |
| Chemical compound, drug | 4–12% Tris-Glycine gel | Invitrogen | Cat#: WXP41220BOX | |
| Chemical compound, drug | PVDF membrane | Millipore | Cat#: ISEQ00010 | |
| Chemical compound, drug | EGTA | Fisher Scientific | Cat#: 28-071G | |
| Chemical compound, drug | Digitonin | Thermo Fisher | Cat#: BN2006 | |
| Chemical compound, drug | G-250 | GoldBio | Cat#: C-460-5 | |
| Chemical compound, drug | 3–12% Bis-Tris Native gel | Invitrogen | Cat#: BN1001BOX | |
| Chemical compound, drug | Acetic acid | Thermo Fisher | Cat#: 9526-33 | |
| Chemical compound, drug | TMRM | Invitrogen | Cat#: I34361 | |
| Chemical compound, drug | JC-10 | AdipoGen | Cat#: 50-114-6552 | |
| Chemical compound, drug | Succinate | Thermo Fisher | Cat#: 041983.A7 | |
| Chemical compound, drug | Hank's Balanced Salt Solution | Thermo Fisher | Cat#: 14025092 | |
| Chemical compound, drug | Calcium Green-5N | Invitrogen | Cat#: C3737 | |

*Appendix 1 Continued*

| Reagent type (species) or resource | Designation | Source or reference | Identifiers | Additional information |
|---|---|---|---|---|
| Chemical compound, drug | Cyclosporine A | Thermo Fisher | Cat#: AC457970010 | |
| Chemical compound, drug | Ethanol | Thermo Fisher | Cat#: R40135 | |
| Chemical compound, drug | Crystal violet | Sigma | Cat#: V5265 | |
| Chemical compound, drug | Proteinase K | Invitrogen | Cat#: 25530049 | |
| Chemical Compound, drug | Caspase-3/7 detection reagents | Invitrogen | Cat#: C10432 | |
| Chemical compound, drug | ATP-red dye | Millipore | Cat#: SCT045 | |
| Chemical compound, drug | MitoTracker-Green | Invitrogen | Cat#: M7514 | |
| Chemical compound, drug | Protein A/G magnetic beads | Pierce | Cat#: 88802 | |
| Chemical compound, drug | M.O.M. blocking reagent | Vector Laboratories | Cat#: BMK-2202 | |
| Software, algorithm | SPSS | SPSS | RRID:SCR_002865 | |
| Software, algorithm | GraphPad Prism 9.4.1 | GraphPad | RRID:SCR_002798 | https://www.graphpad.com/scientific-software/prism/ |
| Software, algorithm | ImageJ 1.53t | NIH | RRID:SCR_003070 | https://imagej.nih.gov/ij/download.html |
| Software, algorithm | ZEN (blue edition) | ZEISS | RRID:SCR_013672 | https://www.zeiss.com/microscopy/us/products/microscope-software.html |
| Software, algorithm | Gen5 | Agilent Technologies (BioTek) | RRID:SCR_017317 | https://www.biotek.com/products/software-robotics-software/gen5-microplate-reader-and-imager-software/ |
| Software, algorithm | Endnote 20 | Clarivate | RRID:SCR_014001 | https://endnote.com/downloads |

