## [Editor Report · eLife Assessment]

Glioblastoma is among the most aggressive cancers without a cure, and its cells are characterized by high mitochondrial membrane potential. This manuscript provides **convincing** evidence that glioblastoma tumorigenesis is closely linked to mitochondrial stress. The study makes a **valuable** contribution to the field by advancing our understanding of the metabolic mechanisms driving glioblastoma and highlighting potential therapeutic targets.

---

## [Referee Report · Reviewer #1 (Public review)]

Summary:

Cai et al have investigated the role of msiCAT-tailed mitochondrial proteins that frequently exist in glioblastoma stem cells. Overexpression of msiCAT-tailed mitochondrial ATP synthase F1 subunit alpha (ATP5) protein increases the mitochondrial membrane potential and blocks mitochondrial permeability transition pore formation/opening. These changes in mitochondrial properties provide resistance to staurosporine (STS)-induced apoptosis in GBM cells. Therefore, msiCAT-tailing can promote cell survival and migration, while genetic and pharmacological inhibition of msiCAT-tailing can prevent the overgrowth of GBM cells.

Strengths:

The CATailing concept has not been explored in cancer settings. Therefore, the present provides new insights for widening the therapeutic avenue.

---

## [Referee Report · Reviewer #2 (Public Review)]

This work explores the connection between glioblastoma, mito-RQC, and msiCAT-tailing. They build upon previous work concluding that ATP5alpha is CAT-tailed and explore how CAT-tailing may affect cell physiology and sensitivity to chemotherapy. The authors conclude that when ATP5alpha is CAT-tailed, it either incorporates into the proton pump or aggregates and that these events dysregulate MPTP opening and mitochondrial membrane potential and that this regulates drug sensitivity. This work includes several intriguing and novel observations connecting cell physiology, RQC, and drug sensitivity. This is also the first time this reviewer has seen an investigation of how a CAT tail may specifically affect the function of a protein.

Comment from the Reviewing Editor:

The revisions made the work more valuable and convincing. The authors adequately made point-by-point response to the reviewers comments by providing new data. Image acquisition and data analysis were further clarified. NEMF knockdown experiments and additional control data for ATP5α featuring a poly-glycine-serine (GS) tail support their conclusion.

---

## [Author Response]

The following is the authors’ response to the previous reviews.

**eLife Assessment:**
Glioblastoma is one of the most aggressive cancers without a cure. Glioblastoma cells are known to have high mitochondrial potential. This useful study demonstrates the critical role of the ribosome-associated quality control (RQC) pathway in regulating mitochondrial membrane potential and glioblastoma growth. Some assays are incomplete; further revision will improve the significance of this study.

For clarity, we propose revising the second sentence to: "It is well-established that certain cancer cells, such as glioblastoma cells, exhibit elevated mitochondrial membrane potential."

**Reviewer #1 (Public Review):**
Summary:Cai et al have investigated the role of msiCAT-tailed mitochondrial proteins that frequently exist in glioblastoma stem cells. Overexpression of msiCAT-tailed mitochondrial ATP synthase F1 subunit alpha (ATP5) protein increases the mitochondrial membrane potential and blocks mitochondrial permeability transition pore formation/opening. These changes in mitochondrial properties provide resistance to staurosporine (STS)-induced apoptosis in GBM cells. Therefore, msiCAT-tailing can promote cell survival and migration, while genetic and pharmacological inhibition of msiCAT-tailing can prevent the overgrowth of GBM cells.Strengths:The CAT-tailing concept has not been explored in cancer settings. Therefore, the present provides new insights for widening the therapeutic avenue.

Your acknowledgment of our study's pioneering elements is greatly appreciated.

Weaknesses:Although the paper does have strengths in principle, the weaknesses of the paper are that these strengths are not directly demonstrated. The conclusions of this paper are mostly well-supported by data, but some aspects of image acquisition and data analysis need to be clarified and extended.

We are grateful for your acknowledgment of our study’s innovative approach and its possible influence on cancer therapy. We sincerely appreciate your valuable feedback. In response, this updated manuscript presents substantial new findings that reinforce our central argument. Moreover, we have broadened our data analysis and interpretation, as well as refined our methodological descriptions.

**Reviewer #2 (Public Review):**
This work explores the connection between glioblastoma, mito-RQC, and msiCAT-tailing. They build upon previous work concluding that ATP5alpha is CAT-tailed and explore how CAT-tailing may affect cell physiology and sensitivity to chemotherapy. The authors conclude that when ATP5alpha is CAT-tailed, it either incorporates into the proton pump or aggregates and that these events dysregulate MPTP opening and mitochondrial membrane potential and that this regulates drug sensitivity. This work includes several intriguing and novel observations connecting cell physiology, RQC, and drug sensitivity. This is also the first time this reviewer has seen an investigation of how a CAT tail may specifically affect the function of a protein. However, some of the conclusions in this work are not well supported. This significantly weakens the work but can be addressed through further experiments or by weakening the text.

We appreciate the recognition of our study's novelty. To address your concerns about our conclusions, we have revised the manuscript. This revision includes new data and corrections of identified issues. Our detailed responses to your specific points are outlined below.

**Reviewer #1 (Recommendations For The Authors):**
(1) In Figure 1B, please replace the high-exposure blots of ATP5 and COX with representative results. The current results are difficult to interpret clearly. Additionally, it would be helpful if the author could explain the nature of the two different bands in NEMF and ANKZF1. Did the authors also examine other RQC factors and mitochondrial ETC proteins? I'm also curious to understand why CAT-tailing is specific to C-I30, ATP5, and COX-V, and why the authors did not show the significance of COX-V.

We appreciate your inquiry regarding the data. Additional attempts were made using new patient-derived samples; however, these results did not improve upon the existing ATP5⍺, (NDUS3)C-I30, and COX4 signals presented in the figure. This is possibly due to the fact that CAT-tail modified mitochondrial proteins represent only a small fraction of the total proteins in these cells. It is acknowledged that the small tails visible above the prominent main bands are not particularly distinct. To address this, the revised version includes updated images to better illustrate the differences. We believe the assertion that GBM/GSCs possess CAT-tailed proteins is substantiated by a combination of subsequent experimental findings. The figure (refer to new Fig. 1B) serves primarily as an introduction. It is important to note that the CAT-tailed ATP5⍺ plays a vital role in modulating mitochondrial potential and glioma phenotypes, a function which has been demonstrated through subsequent experiments.

It is acknowledged that the CAT-tail modification is not exclusive to the ATP5⍺protein. ATP5⍺ was selected as the primary focus of this study due to its prevalence in mitochondria and its specific involvement in cancer development, as noted by Chang YW et al. Future research will explore the possibility of CAT tails on other mitochondrial ETC proteins. Currently, NDUS3 (C-I30), ATP5⍺, and COX4 serve as examples confirming the existence of these modifications. It remains challenging to detect endogenous CAT-tailing, and bulk proteomics is not yet feasible for this purpose. COX4 is considered significant. We hypothesize that CAT-tailed COX4 may function similarly to the previously studied C-I30 (Wu Z, et al), potentially causing substantial mitochondrial proteostasis stress.

Concerning RQC proteins, our blotting analysis of GBM cell lines now includes additional RQC-related factors. The primary, more prominent bands (indicated by arrowheads) are, in our assessment, the intended bands for NEMF and ANKZF1. Subsequent blotting analyses showed only single bands for both ANKZF1 and NEMF, respectively. The additional, larger molecular weight band of NEMF, which was initially considered for property analysis (phosphorylation, ubiquitination, etc.), was not examined further as it did not appear in subsequent experiments (refer to new Fig. S1C).

References:

Chang YW, et al. Spatial and temporal dynamics of ATP synthase from mitochondria toward the cell surface. Communications biology. 2023;6(1).

Wu Z, et al. MISTERMINATE Mechanistically Links Mitochondrial Dysfunction With Proteostasis Failure. Molecular cell. 2019;75(4).

(2) In addition to Figure 1B, it would be interesting to explore CAT-tailed mETC proteins in cancer tissue samples.

This is an excellent point, and we appreciate the question. We conducted staining for ATP5⍺ and key RQC proteins in both tumor and normal mouse tissues. Notably, ATP5⍺ in GBM exhibited a greater tendency to form clustered punctate patterns compared to normal brain tissue, and not all of it co-localized with the mitochondrial marker TOM20 (refer to new Fig. S3C-E). Crucially, we observed a significant increase in NEMF expression within mouse xenograft tumor tissues, alongside a decrease in ANKZF1 expression (refer to new Fig. S1A, B). These findings align with our observations in human samples.

(3) Please knock down ATP5 in the patient's cells and check whether both the upper band and lower band of ATP5 have disappeared or not.

This control was essential and has been executed now. To validate the antibody's specificity, siRNA knockdown was performed. The simultaneous elimination of both upper and lower bands upon siRNA treatment (refer to new Fig. S2A) confirms they represent genuine signals recognized by the antibody.

(4) In Figure 1C and ID, add long exposure to spot aggregation and oligomer. Figure 1D, please add the blots where control and ATP5 are also shown in NHA and SF (similar to SVG and GSC827).

New data are included in the revised manuscript to address the queries. Specifically, the new Fig 1D now displays the full queue as requested, featuring blots for Control, ATP5α, AT3, and AT20. Our analysis reveals that AT20 aggregates exhibit higher expression and accumulation rates in GSC and SF cells.

Fig. 1C has been updated to include experimental groups treated with cycloheximide and sgNEMF. Our results show that sgNEMF effectively inhibits CAT-tailing in GBM cell lines, whereas cycloheximide has no impact. After consulting with the Reporter's original creator and optimizing expression conditions, we observed no significant aggregates with β-globin-non-stop protein, potentially due to the length of endogenous CAT-tail formation (as noted by Inada, 2020, in Cell Reports). Our analysis focused on the ratio of CAT-tailed (red box blots) and non-CAT-tailed proteins (green box blots). Comparing these ratios revealed that both anisomycin treatment and sgNEMF effectively hinder the CAT-tailing process, while cycloheximide has no effect.

(5) In Figure 1E, please double-check the results with the figure legend. ATP5A aggregated should be shown endogenously. The number of aggregates shown in the bar graph is not represented in micrographs. Please replace the images. For Figure 1E, to confirm the ATP5-specific aggregates, it would be better if the authors would show endogenous immunostaining of C-130 and Cox-IV.

Labels in Fig. 1E were corrected to reflect that the bar graph in Fig. 1F indicates the number of cells with aggregates, not the quantity of aggregates per cell. The presence

(6) Figure 3A. Please add representative images in the anisomycin sections. It is difficult to address the difference.

We appreciate your feedback. Upon re-examining the Calcein fluorescence intensity data in Fig. 3A, we believe the images accurately represent the statistical variations presented in Fig. 3B. To address your concerns more effectively, please specify which signals in Fig. 3A you find potentially misleading. We are prepared to revise or substitute those images accordingly.

(7) Figure 3D. If NEMF is overexpressed, is the CAT-tailing of ATP 5 reversed?

Thank you. Your prediction aligns with our findings. We've added data to the revised Fig. S6A, B, which demonstrates that both NEMF overexpression and ANKZF1 knockdown lead to elevated levels of CRC. This increase, however, was not statistically significant in GSC cells. A plausible explanation for this discrepancy is that the MPTP of GSC cells is already closed, thus any additional increase in CAT-tailing activity does not result in further amplification.

(8) Figure 3G. Why on the BN page are AT20 aggregates not the same as shown in Figure 2E?

We appreciate your inquiry regarding the ATP5⍺ blots, specifically those in the original Fig. 3G (left) and 2E (right). Careful observation of the ATP5⍺ band placement in these figures reveals a high degree of similarity. Notably, there are aggregates present at the top, and the diffuse signals extend downwards. Given that this is a gradient polyacrylamide native PAGE, the concentration diminishes towards the top. Consequently, the non-rigid nature of the Blue Native PAGE gel may lead to slight variations in the aggregate signals; however, the overall patterns are very much alike. To mitigate potential misinterpretations, we have rearranged the blot order in the new Fig. 3M.

(9) Figure 4D. The amount of aggregation mediated by AT20 is more compared to AT3. Why are there no such drastic effects observed between AT3 and AT20 in the Tunnel assay?

The previous Figure 4D presents the quantification of cell migration from the experiment depicted in Figure 4C. But this is a good point. TUNEL staining results are directly influenced by mitochondrial membrane potential and the state of mitochondrial permeability transition pores

(MPTP), not by the degree of protein aggregation. Our previous experiments showed comparable effects of AT3 and AT20 on mitochondria (Fig. 2E, 3K), which aligns with the expected similar outcomes on TUNEL staining. As for its biological nature, this could be very complicated. We hope to explore it in future studies.

(10) Figure 5C: The role of NEMF and ANKZF1 can be further clarified by conducting Annexin-PI assays using FACS. The inclusion of these additional data points will provide more robust evidence for CAT-tailing's role in cancer cells.

In response to your suggestion, we have incorporated additional data into the revised version.Using the Annexin-PI kit, we labeled apoptotic cells and detected them using flow cytometry (FACS). Our findings indicate that anisomycin pretreatment, NEMF knockdown (sgNEMF), and ANZKF1 upregulation (oeANKZF1) significantly increase the rate of STS-induced apoptosis compared to the control group (refer to new Fig. S9D-G).

(11) Figure 5F: STS is a known apoptosis inhibitor. Why it is not showing PARP cleavage? Also, cell death analysis would be more pronounced, if it could be shown at a later time point. What is the STS and Anisomycin at 24h or 48h time-point? Since PARP is cleaved, it would also be better if the authors could include caspase blots.

I guess what you meant to say here is "Staurosporine is a protein kinase inhibitor that can induce apoptosis in multiple mammalian cell lines." Our study observed PARP cleavage even in GSCs, which are typically more resistant to staurosporine-induced apoptosis (C-PARP in Fig. S9B). The ratio of C-PARP to total PARP increased. We selected a 180-minute treatment duration because longer treatments with STS + anisomycin led to a late stage of apoptosis and non-specific protein degradation (e.g., at 24 or 48 hours), making PARP comparisons less meaningful. Following your suggestion, we also examined caspase 3/7 activity in GSC cells treated with DMSO, CHX, and anisomycin. We found that anisomycin treatment also activated caspases (Fig. S9A).

(12) In Figure 5, the addition of an explanation, how CAT-tailing can induce cell death, would add more information such as BAX-BCL2 ratio, and cytochrome-c release from the mitochondria.

Thank you for your suggestion. In this study, we state that specific CAT-tails inhibit GSC cell death/apoptosis rather than inducing it. Therefore, we do not expect that examining BAX-BCL2 and mitochondrial cytochrome c release would offer additional insights.

(13) To confirm the STS resistance, it would be better if the author could do the experiments in the STS-resistant cell line and then perform the Anisomycin experiments.

Thank you. We should emphasize that our data primarily originates from GSC cells. These cells already exhibit STS-resistance when compared to the control cells (Fig. S8A-C).

(14) It would be more advantageous if the author could show ATP5 CATailed status under standard chemotherapy conditions in either cell lines or in vivo conditions.

This is an interesting question. It's worth exploring this question; however, GSC cells exhibit strong resistance to standard chemotherapy treatments like temozolomide (TMZ).

Additionally, we couldn't detect changes in CAT-tailed ATP5⍺ and thus did not include that data.

(15) In vivo (cancer mouse model or cancer fly model) data will add more weight to the story.

We appreciate your intriguing question. An effective approach would be to test the RQC pathway's function using the Drosophila Notch overexpression-induced brain tumor model. However, Khaket et al. have conducted similar studies, stating, "The RNAi of Clbn, VCP, and Listerin (Ltn), homologs of key components of the yeast RQC machinery, all attenuated NSC over-proliferation induced by Notch OE (Figs. 5A and S5A–D, G)." This data supports our theory, and we have incorporated it into the Discussion. While the mouse model more closely resembles the clinical setting, it is not covered by our current IACUC proposal. We intend to verify this hypothesis in a future study.

Reference:

Khaket TP, Rimal S, Wang X, Bhurtel S, Wu YC, Lu B. Ribosome stalling during c-myc translation presents actionable cancer cell vulnerability. PNAS Nexus. 2024 Aug 13;3(8):pgae321.

**Reviewer #2 (Recommendations For The Authors):**
Figure 1B, C: To demonstrate that Globin, ATP5alpha, and C-130 are CAT-tailed, it is necessary to show that the high mobility band disappears after NEMF deletion or mutagenesis of the NFACT domain of NEMF. This can be done in a cell line. The anisomycin experiment is not convincing because the intensity of the bands drops and because no control is done to show that the effects are not due to translation inhibition (e.g. cycloheximide, which inhibits translation but not CAT tailing). Establishing ATP5alpha as a bonafide RQC substrate and CAT-tailed protein is critical to the relevance of the rest of the paper.

Thank you for suggesting this crucial control experiment. To confirm the observed signal is indeed a bona fide CAT-tail, it's essential to demonstrate that NEMF is necessary for the CAT-tailing process. We have incorporated data from NEMF knockdown (sgNEMF) and cycloheximide treatment into the revised manuscript. Our findings show that both sgNEMF and anisomycin treatment effectively inhibit the formation of CAT-tailing signals on the reporter protein (Fig. 1C). Similarly, NEMF knockdown in a GSC cell line also effectively eliminated CAT-tails on overexpressed ATP5⍺ (Fig. S2B).

In general, the text should be weakened to reflect that conclusions were largely gleaned from artificial CAT tails made of AT repeats rather than endogenously CAT-tailed ATP5alpha. CAT tails could have other sequences or be made of pure alanine, as has been suggested by some studies.

Thank you for your reminder. We have reviewed the recent studies by Khan et al. and Chang et al., and we found their analysis of CAT tail components to be highly insightful. We concur with your suggestion regarding the design of the CAT tail sequence. We aimed to design a tail that maintained stability and resisted rapid degradation, regardless of its length. In the revised version, we clarify that our conclusions are based on artificial CAT tails, specifically those composed of AT repeat sequences (p. 9). We acknowledge that the presence of other sequence components may lead to different outcomes (p. 19).

Reference:

Khan D, Vinayak AA, Sitron CS, Brandman O. Mechanochemical forces regulate the composition and fate of stalled nascent chains. bioRxiv [Preprint]. 2024 Oct 14:2024.08.02.606406. Chang WD, Yoon MJ, Yeo KH, Choe YJ. Threonine-rich carboxyl-terminal extension drives aggregation of stalled polypeptides. Mol Cell. 2024 Nov 21;84(22):4334-4349.e7.

Throughout the work (e.g. 3B, C), anisomycin effects should be compared to those with cycloheximide to observe if the effects are specific to a CAT tail inhibitor rather than a translation inhibitor.

We agree that including cycloheximide control experiments is crucial. The revised version now incorporates new data, as depicted in Fig. S5A, B, illustrating alterations in the on/off state of MPTP following cycloheximide treatment. Furthermore, Fig. S6A, B present changes in Calcium Retention Capacity (CRC) under cycloheximide treatment. The consistency of results across these experiments, despite cycloheximide treatment, suggests that anisomycin's role is specifically as a CAT tail inhibitor, rather than a translation inhibitor.

Line 110, it is unclear what "short-tailed ATP5" is. Do you mean ATP5alpha-AT3? If so this needs to be introduced properly. Line 132: should say "may indicate accumulation of CAT-tailed protein" rather than "imply".

We acknowledge your points. We have clarified that the "short-tailed ATP5α" refers to ATP5α-AT3 and incorporated the requested changes into the revised manuscript.

Figure 1C: how big are those potential CAT-tails (need to be verified as mentioned earlier)?They look gigantic. Include a ladder.

In the revised Fig. 1D, molecular weight markers have been included to denote signal sizes. The aggregates in the previous Fig. 1C, also present in the control plasmid, are likely a result of signal overexposure. The CAT-tailed protein is observed just above the intended band in these blots. These aggregates have been re-presented in the updated figures, and their signal intensities quantified.

Line 170: "indicating that GBM cells have more capability to deal with protein aggregation". This logic is unclear. Please explain.

We appreciate your question and have thoroughly re-evaluated our conclusion. We offer several potential explanations for the data presented in Fig. 1D: (1) ATP5α-AT20 may demonstrate superior stability. (2) GSC (GBM) cells might lack adequate mechanisms to monitor protein accumulation. (3) GSC (GBM) cells could possess an increased adaptive capacity to the toxicity arising from protein accumulation. This discussion has been incorporated into the revised manuscript (lines 166-169).

Line 177: how do you know the endogenous ATP5alpha forms aggregates due to CAT-tailing? Need to measure in a NEMF hypomorph.

We understand your concern and have addressed it. Revised Fig. 3G, H demonstrates that a reduction in NEMF levels, achieved through sgNEMF in GSC cells, significantly diminishes ATP5α aggregation. This, in conjunction with the Anisomycin treatment data presented in revised Fig. 3E, F, confirms the substantial impact of the CAT-tailing process on this aggregation.

Line 218: really need a cycloheximide or NEMF hypomorph control to show this specific to CAT-tailing.

We have revised the manuscript to include data from sgNEMF and cycloheximide treatments, specifically Fig. 3G, H, and Fig. S5C, D, as detailed in our response above.

Lines 249,266, Figure 5A: The mentioned experiments would benefit from controls including an extension of ATP5alpha that was not alanine and threonine, perhaps a gly-ser linker, as well as an NEMF hypomorph.

We sincerely appreciate your insightful comments. In response, the revised manuscript now incorporates control data for ATP5α featuring a poly-glycine-serine (GS) tail. This data is specifically presented in Figs. S2E-G, S4E, S7A, D, E, and S8F, G. Our experimental findings consistently demonstrate that the overexpression of ATP5α, when modified with GS tails, had no discernible impact on protein aggregation, mitochondrial membrane potential, GSC cell mobility, or any other indicators assessed in our study.

Figure S5A should be part of the main figures and not in the supplement.

This has been moved to the main figure (Fig. 5C).